# Non-viral generation of transgenic non-human primates via the piggyBac transposon system

Masataka Nakaya[1,2], Chizuru Iwatani[1], Setsuko Tsukiyama-Fujii[1], Ai Mieda[2], Shoko Tarumoto[2], Taro Tsujimura[2], Takuya Yamamoto [2,3,4,5], Takafumi Ichikawa [2,6], Tomonori Nakamura [2,7,8], Ichiro Terakado[1], Ikuo Kawamoto[1], Takahiro Nakagawa[1], Iori Itagaki[1,9], Mitinori Saitou [2,3,7], Hideaki Tsuchiya[1] & Tomoyuki Tsukiyama [1,2] ✉

Non-human primates, such as cynomolgus monkeys, are invaluable experimental models for understanding human biology and disease. Their close genetic relationship to humans makes them essential for studying fundamental human developmental processes and disease progression. Although lentiviral methods for generating transgenic monkeys exist, several inherent technical difficulties limit their utility. To solve this problem, here we establish a non-viral method for generating transgenic cynomolgus monkeys using the piggyBac transposon system. After optimizing our protocol in mice, we show that the co-injection of piggyBac components with sperm into metaphase II-stage oocytes successfully generates transgenic monkeys expressing transgenes throughout their whole bodies. Transgene expression is observed in all examined tissue types, including germ cells, although the levels of expression vary. Insertion analysis further confirms the successful integration of the transgene. We propose that our method will be a practical non-viral protocol for generating transgenic non-human primates.

Genetic engineering of experimental animal models has contributed significantly to our current understanding in the fields of biology and medicine, including by elucidating gene functions and recapitulating different aspects of human diseases[1–3]. The ability to generate transgenic mice has been instrumental in this advance. However, one major caveat of small animal model systems is that they rarely reflect actual human physiology and metabolic functions, which makes accurately reproducing human disease pathology difficult[4]. Therefore, it is highly desirable to establish an animal model system that more closely resembles human development and disease[5]. Non-human primates,

being the closest relatives to humans, have served as an invaluable experimental model system for elucidating not only the basic principles of what makes us human but also fundamental processes of development and disease progression[3,6,7].

Cynomolgus monkeys, among the non-human primate species that can be experimentally manipulated, are the closest to humans and are widely used in neuroscience and infectious disease research, safety evaluations in drug development, and preclinical studies[7]. We have thus far generated multiple types of genetically modified cynomolgus monkeys, including transgenic monkeys using the lentiviral vector

[1]Research Center for Animal Life Science, Shiga University of Medical Science, Shiga, Japan. [2]Institute for the Advanced Study of Human Biology (WPI-ASHBi), Kyoto University, Kyoto, Japan. [3]Center for iPS Cell Research and Application (CiRA), Kyoto University, Kyoto, Japan. [4]AMED-CREST, AMED, Tokyo, Japan. [5]Medical-Risk Avoidance Based on iPS Cells Team, RIKEN Center for Advanced Intelligence Project (AIP), Kyoto, Japan. [6]Department of Developmental Biology, Graduate School of Medicine, Kyoto University, Kyoto, Japan. [7]Department of Anatomy and Cell Biology, Graduate School of Medicine, Kyoto University, Kyoto, Japan. [8]The HAKUBI center for Advanced Research, Kyoto University, Kyoto, Japan. [9]The Corporation for Production and Research of Laboratory Primates, Ibaraki, Japan. ✉e-mail: ttsuki@belle.shiga-med.ac.jp

method and gene knock-out monkeys using the CRISPR/Cas9 method[8-11]. For producing transgenic monkeys, the lentiviral vector method has been employed to overcome the shortcomings of conventional pronuclear (PN) injection methods. PN injection requires the manipulation of many embryos and the generation of multiple lines due to potential transgene truncations and genetic mosaicism, making it a highly inefficient and impractical method to generate transgenic monkeys in terms of time, cost, and labor.

Alternatively, viral-mediated transgenesis in monkeys offers a more robust approach than PN injection[9,11-16]. However, the lentiviral vector has also been associated with several technical difficulties. First, viral methods require special equipment, facilities and skills, limiting their utility. Additionally, the lentiviral method has several methodological shortcomings, such as limitation in screening for transgene integration prior to recipient transfer[10,11], limits to the length of transgene insert[17], and genetic mosaicism[17]. Among them, one major technical difficulty is the preimplantation screening. The selection of correctly modified embryos before their transfer to recipient mothers is critically important because experiments in cynomolgus monkeys require ethical prudence and sacrificing monkeys solely due to improper genetic modifications is ethically unacceptable[18]. However, in the case of lentiviral transgenic monkey generation, some of the resultant monkeys delivered did not show clear fluorescence even though we confirmed the fluorescence before transferring embryos[9-11]. We attributed this lack of clear fluorescence to lentiviral-production-derived debris exhibiting fluorescence and hindering the evaluation of transgene expression[10]. This problem can arise when using lentiviral vectors that encode fluorescent proteins, since the fluorescent proteins will also be produced in the virus packaging cells during the lentiviral particle production process. Even after purification of the lentiviral particles, the fluorescent proteins can be carried over and contaminate the viral purification solution. Hence, there is a critical need for a better, more robust, and versatile approach to generate transgenic monkeys.

To overcome the current limitations of transgenesis in monkeys, here we develop a non-viral method, using the piggyBac transposon system, to generate a transgenic monkey. The piggyBac transposon system allows the incorporation of large genomic sequences[19,20], but also does not require special equipment or facilities for viral production, making it a more simple and versatile strategy to generate transgenic animals than existing viral methods. Importantly, our approach allows for the easy selection of embryos with transgene expression before the transfer to recipient mothers because, unlike in the case of viral methods, no fluorescence debris is present. Therefore, it also prevents the unnecessary sacrificing of monkeys, making it more ethically acceptable than other existing conventional approaches. However, previous reports on the use of the piggyBac system in non-human primates have been limited to embryonic experiments, and whether this system can generate live transgenic non-human primates, including cynomolgus monkeys, remains unknown[18,21]. In the present study, to produce transgenic monkeys using piggyBac, we optimize the conditions of piggyBac injection for the generation of transgenic cynomolgus monkeys.

## Results

### Determining the optimal conditions for piggyBac co-injection with sperm in mice

In contrast to mice, for which various methods can be used to obtain embryos, including mating, in vitro fertilization (IVF), and intracytoplasmic sperm injection (ICSI), only ICSI is practical for obtaining embryos in monkeys because of cost and reliability[22]. Thus, the conventional PN injection method in monkeys must be used in conjunction with ICSI and requires multiple embryo manipulation. We speculated that co-injection of piggyBac components during ICSI could reduce damage to the embryos and also be more efficient in

producing transgenic monkeys. To determine the optimal conditions for transgenesis, we first co-injected piggyBac components with sperm into metaphase II (MII)-stage oocytes in mice and compared this approach with the PN injection method (Fig. 1a). Given the substantial cost and ethical considerations associated with primate research, these comparisons were conducted in mice. We used a piggyBac vector-containing membrane tdTomato and histone 2B (H2B) green fluorescent protein (GFP) under control of the human cytomegalovirus immediate-early enhancer/chicken beta-actin (CAG) promoter (Fig. 1b). The membrane tdTomato was used to distinguish the autofluorescence in live imaging as monkey blastocyst embryos are empirically known to exhibit green nuclear autofluorescence (Supplementary Fig. 1). H2B-GFP was used to evaluate the positive expression rate after fixation because the autofluorescence disappears after fixation.

We compared the transgene expression at the blastocyst stage between PN injection and co-injection (Fig. 1c, d). Because transient expression from the vector before genome insertion—which is diluted out as cell division proceeds—can affect the evaluation for the expression of the inserted transgenes, we investigated the optimal concentration of the piggyBac vector under conditions with or without the piggyBac transposase (PBase). Using PN injection, 5 ng/μL was found to be the optimal concentration (Fig. 1e, f). Injection of more than 5 ng/μL caused a decrease in GFP-positive cell rates and GFP intensities. Under these conditions, most embryos could not develop to the blastocyst stage and only GFP-negative or -weakly positive blastocysts were obtained, suggesting that PN injection of a high concentration of the piggyBac vector was toxic and adversely affected the developmental ability of the embryos (Fig. 1e and Supplementary Fig. 2). However, upon co-injection, a concentration of 5 ng/μL or less was insufficient for clear expression of the transgenes (Fig. 1g, h). In addition, 30 ng/μL of vector injection without the PBase resulted in residual transient expression at the blastocyst stage (Fig. 1d, g, h). Consequently, we determined that 10 ng/μL was the optimal concentration for co-injection (Fig. 1d, g, h). The developmental rates of the injected embryos are shown in Supplementary Fig. 2.

### Generation of transgenic mice using the co-injection method

To examine whether the co-injection method can generate transgenic mice, we co-injected 10 ng/μL of the piggyBac vector and PBase mRNA into mouse MII oocytes with sperm. The resultant transgenic embryos developed to the blastocyst stage, and transgene expression was determined using membrane tdTomato (Fig. 2a). In total, among 21 blastocysts, 13 membrane tdTomato-positive embryos were selected and transferred into the uterus of recipient mother mice (Fig. 2b). Seven fetuses were delivered, and all of them were transgenic mice, as judged by the fluorescence expressions (Fig. 1b, c and Supplementary Fig. 3). In addition, the presence of transgenes was confirmed by genomic polymerase chain reaction (PCR) (Fig. 2d). These results demonstrate that co-injection can generate transgenic mice and that the evaluation before embryo transfer is useful for efficient derivation of transgenic mice.

F1 mice generated by mating a founder (F0) male with C57BL/6 females exhibited a transgene transmission rate of 72.2% (13/18, Fig. 2e). One F1 individual carried EGFP but not tdTomato. Repeat PCR confirmed the presence of EGFP, at detection levels comparable to those of other positive individuals, suggesting that contamination was unlikely. The disparate result was thus likely due to random integration of a partial vector sequence. Nevertheless, this is an infrequent phenomenon (1/13) that does not compromise the overall practicality of the system. Additionally, ICSI experiments using sperm from three different F1 male mice resulted in high fluorescence-positive embryo rates of 85.7%, 72.7%, and 54.5%, demonstrating efficient germline transmission (Fig. 2f). These results support the robustness and practicality of the piggyBac system for generating transgenic lines.

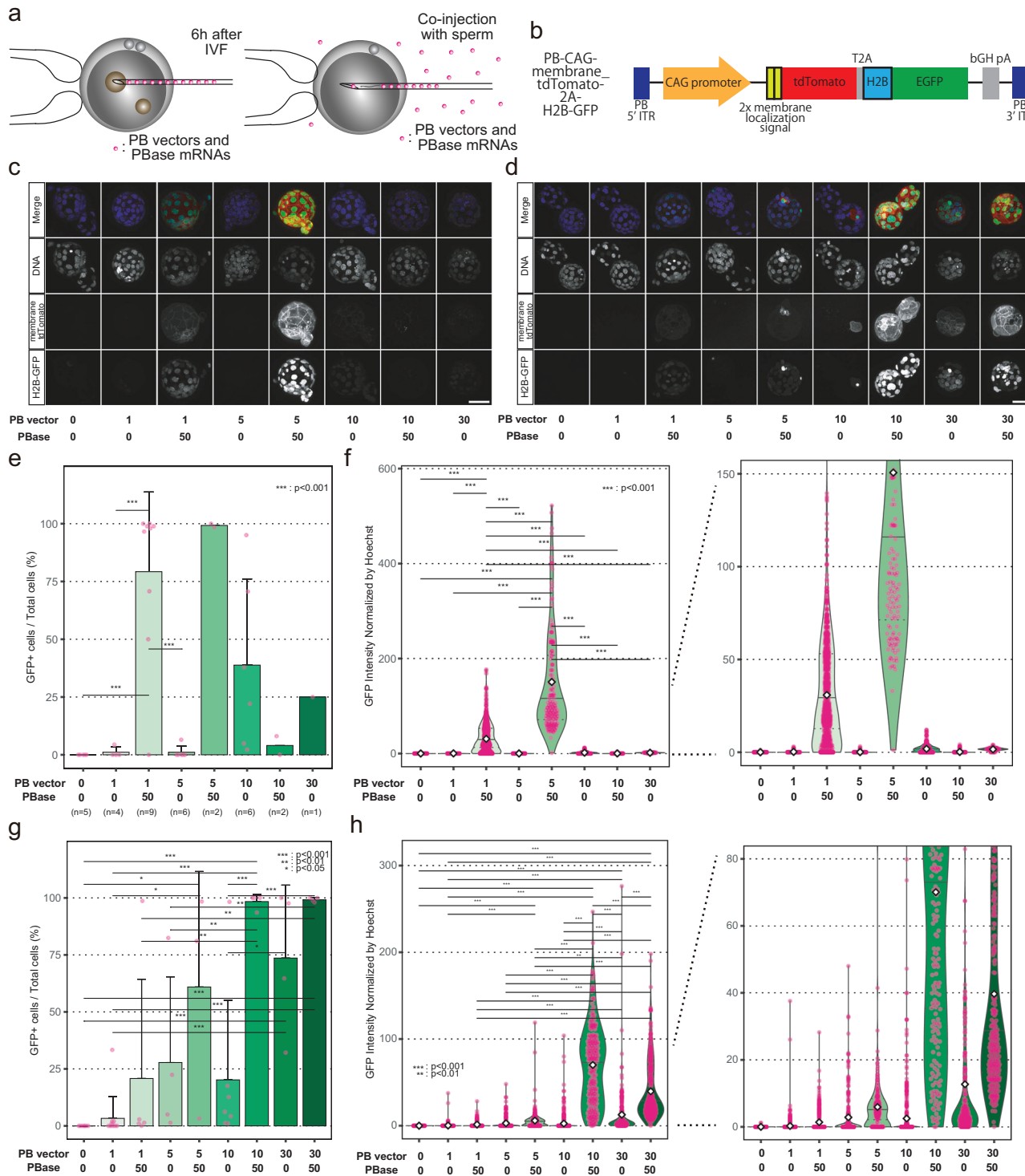

**Fig. 1 | Identification of the optimized conditions for piggyBac co-injection in mice. a** Schematic of the PN injection and co-injection with sperm. "PB" indicates piggyBac. **b** Schematic of the piggyBac vector. **c** Expression of fluorescence reporters in the blastocyst embryos after PN injection. The numbers below each panel are given in ng/μL. Scale bar, 50 μm. **d** Expression of fluorescence reporters in the blastocyst embryos after co-injection. Scale bar, 50 μm. **e** Bar graph of GFP-positive rates in each PN-injected embryo. Error bars, mean values + s.d. In total, there were $n = 35$ biologically independent samples. One-way ANOVA and Tukey-Kramer post-hoc contrasts were used for the comparisons. ***$P < 0.001$. **f** Violin plots of GFP intensities in each nucleus of PN-injected embryos. The top and bottom edges of the violins indicate the maximum and minimum values, respectively;

the center lines indicate the medians; and the dotted lines indicate the first and third quartiles, respectively; the diamonds are the mean values. One-way ANOVA and Tukey-Kramer post-hoc contrasts were used for the comparisons. ***$P < 0.001$. **g** Bar graph of GFP-positive rates in each ICSI co-injected embryo. Error bars, mean values + s.d. In total, there were $n = 52$ biologically independent samples. One-way ANOVA and Tukey-Kramer post-hoc contrasts were used for the comparisons. *$P < 0.05$; **$P < 0.01$; ***$P < 0.001$. **h** Violin plots of GFP intensities in each nucleus of ICSI co-injected embryos. One-way ANOVA and Tukey-Kramer post-hoc contrasts were used for the comparisons. **$P < 0.01$; ***$P < 0.001$. Source data and $P$ values are provided as a Source Data file.

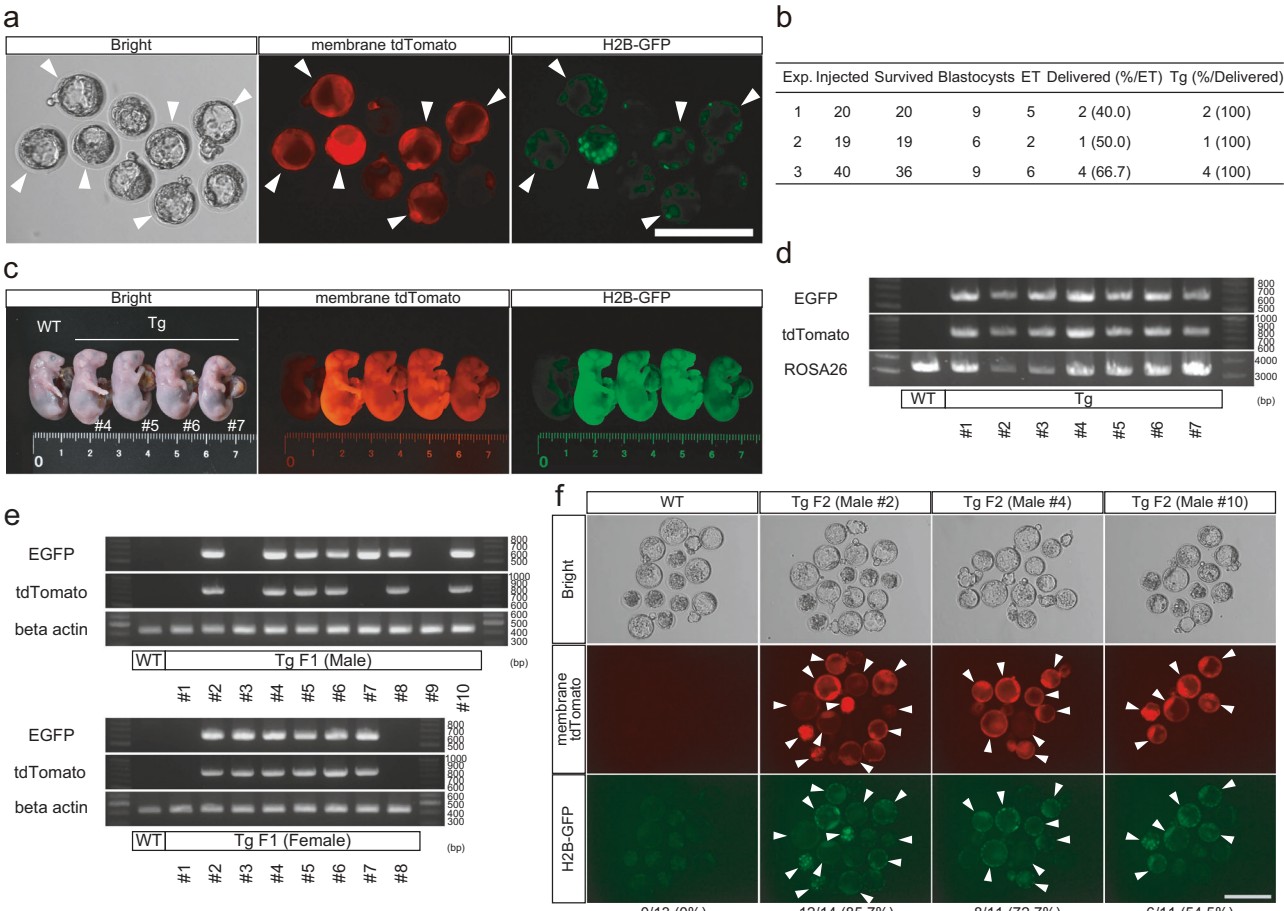

**Fig. 2 | Generation of transgenic mice using the co-injection method.**
**a** Expressions of fluorescence reporters in the blastocyst embryos after co-injection. Scale bar, 200 μm. **b** Efficiencies of fluorescence positive rates in the offspring. "ET" indicates embryo transfer. **c** Expressions of the fluorescence reporters in the offspring. **d** The genomic PCR for the fluorescence reporters in the offspring. **e** The genomic PCR for the fluorescence reporters in the F1 offspring. **f** Expressions of fluorescence reporters in the F2 blastocyst embryos. Source data are provided as a Source Data file.

## Generation of transgenic monkeys using co-injection

As a next step, to examine whether similar results could be obtained in monkeys, we co-injected various concentrations of the piggyBac vector and PBase mRNA into monkey MII oocytes with sperm. We found that, as in mice, 10 ng/μL was the optimal concentration for co-injection in monkeys (Fig. 3a, b, c and Supplementary Fig. 4).

To generate transgenic embryos for embryo transfer, we used the optimal concentration. In total, among 34 blastocysts, 32 embryos were positive for membrane tdTomato (Fig. 4a). The majority exhibited higher mean and median intensities compared to the control group, whereas in some embryos only the maximum intensities exceeded the upper limit of the control, suggesting partial positivity (Fig. 4b, c). Among these tdTomato-positive embryos, 15 embryos were genotyped for sex determination to facilitate efficient production of the next generation through the selection of males. To achieve this, the zona pellucida of transgenic embryos was perforated with a laser at the four-cell stage (Fig. 4d). This procedure facilitated hatching of the embryos at the blastocyst stage. A portion of the hatched cells were sampled, and the genomic DNAs were collected (Fig. 4d). Genotyping of these embryos revealed that 8 were male. These male embryos were then transferred directly without freeze-thawing, whereas the remaining embryos underwent freeze-thawing before being transferred (Fig. 4a). Although several embryos died following the sampling or freeze-thawing, a total of 20 embryos, including the 8 male embryos, were ultimately transferred into recipient mother monkeys (Fig. 4a, b). Each embryo was transferred into an individual recipient

female. As a result of these experiments, one male monkey was delivered from a genotyped embryo (#1, Fig. 4a). This monkey exhibited the fluorescence of tdTomato and GFP in both the placenta and the whole body (Fig. 4e). Additionally, 2 female monkeys (#2 and #3) were delivered, and 1 male monkey (#4) was stillborn (Fig. 4a). Among these monkeys, #2 and #4 demonstrated tdTomato and GFP fluorescence throughout their bodies, whereas the fluorescence of #3 was not apparent (Fig. 4f, g). These results were consistent with the fluorescence checks performed on the preimplantation embryos (Fig. 4c).

## Detection of the transgenes in the tissues of transgenic monkeys

Unfortunately, the delivered male monkey (#1) died from child neglect by the mother monkey. To evaluate tissue-specific variations in transgene expression among transgenic monkeys generated using this experimental system, we investigated the organs of the deceased monkey (#1) and the stillborn monkey (#4). Comprehensive necropsy examinations were performed on both monkeys, and no pathological abnormalities were observed. Genomic PCR revealed the presence of transgenes in all analyzed tissues of these monkeys (Fig. 5a). Most of the organs showed the fluorescence of tdTomato and GFP (Supplementary Figs. 5 and 6). Quantitative RT-PCR showed that all the tissues from monkey #1 expressed GFP, although the expression levels varied among tissues (Fig. 5b). Additionally, Western blot analysis was conducted to independently confirm the transgene expression, and the results showed that GFP protein was expressed in all tissues analyzed (Fig. 5c and Supplementary Fig. 7).

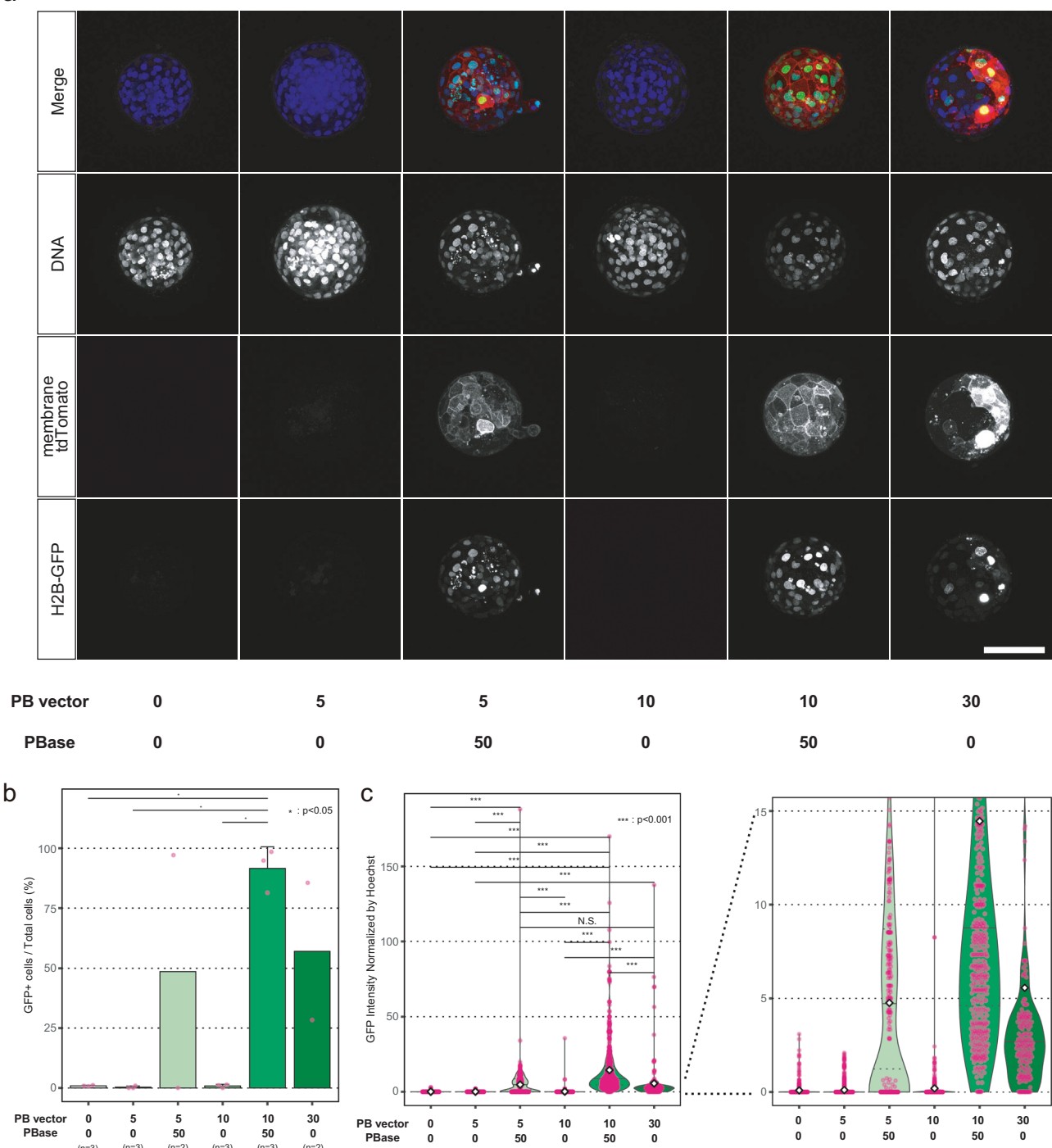

**Fig. 3 | Identification of the optimized conditions for piggyBac co-injection in monkeys. a** Expressions of the fluorescence reporters in the blastocyst embryos after co-injection. "PB" indicates piggyBac. The numbers below each panel are given in ng/μL. Scale bar, 100 μm. **b** Bar graph of GFP-positive rates in each co-injected embryo. Error bars, mean values + s.d. In total, there were $n = 16$ biologically independent samples. One-way ANOVA and Tukey-Kramer post-hoc contrasts were used for the comparisons. *$P < 0.05$. **c** Violin plots of GFP intensities in each nucleus of ICSI co-injected embryos. The top and bottom edges of the violins indicate the maximum and minimum values, respectively; the center lines indicate the medians; and the dotted lines indicate the first and third quartiles, respectively; the diamonds are the mean values. One-way ANOVA and Tukey-Kramer post-hoc contrasts were used for the comparisons. ***$P < 0.001$; N.S. not significant. Source data and $P$ values are provided as a Source Data file.

To detect the transgenes in the live monkeys (#2 and #3), we conducted droplet digital PCR (ddPCR), a highly sensitive method that facilitates the detection of transgenes from minimal sample volumes. This assay clearly demonstrated that the placenta and hair of monkey #2 were positive for EGFP, tdTomato and 5′ terminal repeat (5′ TR) (Fig. 5d). Additionally, the placenta of monkey #3 was positive for these markers and its hair was positive for EGFP. However, the signals of tdTomato and 5′ TR were not apparent in the hair of monkey #3 (Fig. 5d). To verify these results, we also assayed blood samples using the same method. In this assay, no tdTomato or 5′ TR signals were

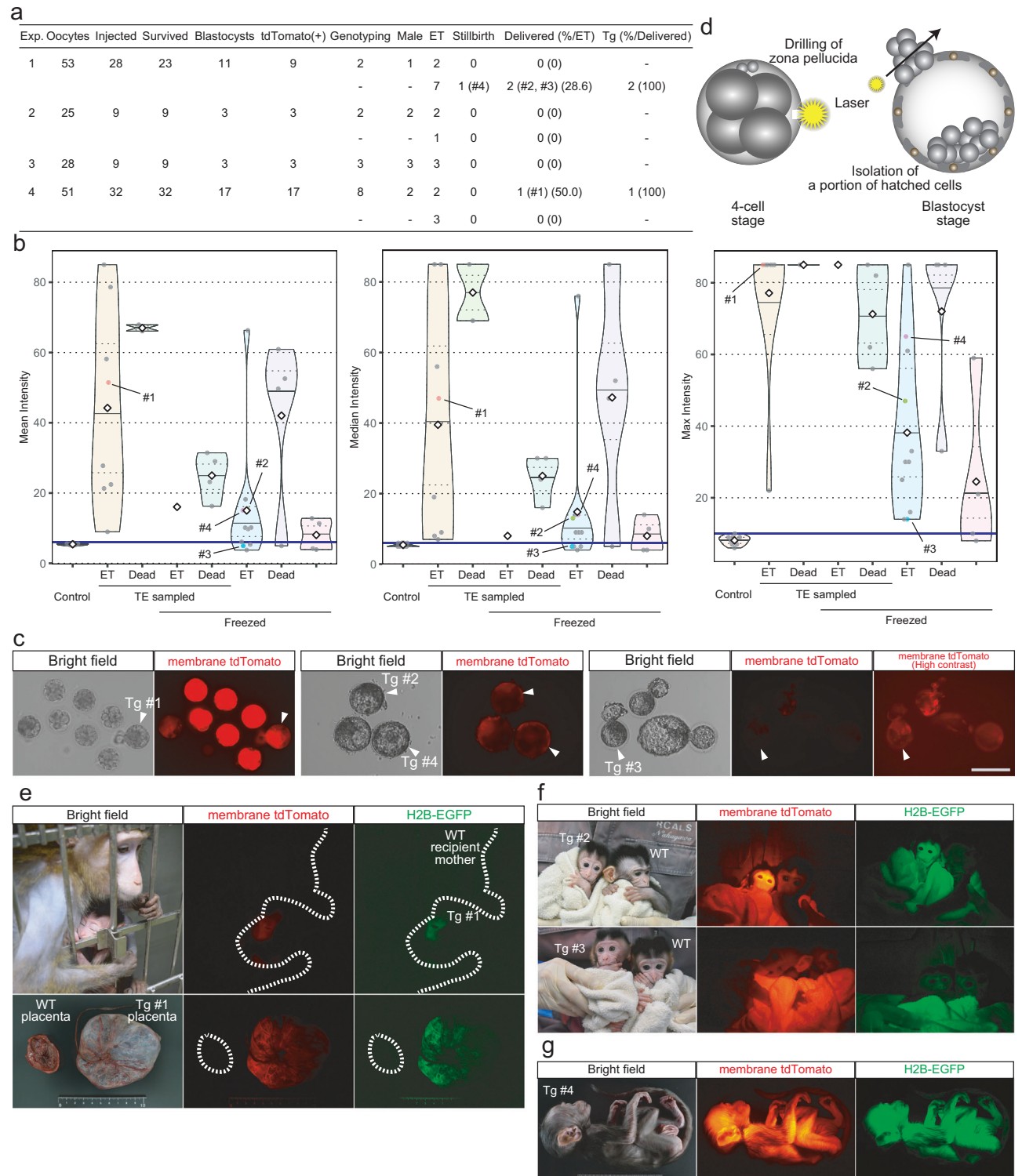

**Fig. 4 | Generation of transgenic monkeys using the co-injection method. a** The numbers of embryos at each step and delivered monkeys. One monkey was used per experiment. "ET" indicates embryo transfer. **b** Violin plots representing the mean, median and maximum tdTomato fluorescence intensities for each blastocyst embryo. The top and bottom edges of the violins indicate the maximum and minimum values, respectively; the center lines indicate the medians; and the dotted lines indicate the first and third quartiles, respectively; the diamonds indicate the mean values. **c** Expressions of tdTomato in the blastocyst embryos after co-injection. Scale bar, 200 μm. **d** Schematic of the preimplantation genotyping with laser-assisted hatching. **e** Expressions of fluorescence reporters in a delivered monkey (#1) and the placenta. **f** Expressions of fluorescence reporters in the delivered monkeys (#2 and #3). **g** Expression of fluorescence reporters in the stillborn monkey (#4). Source data are provided as a Source Data file.

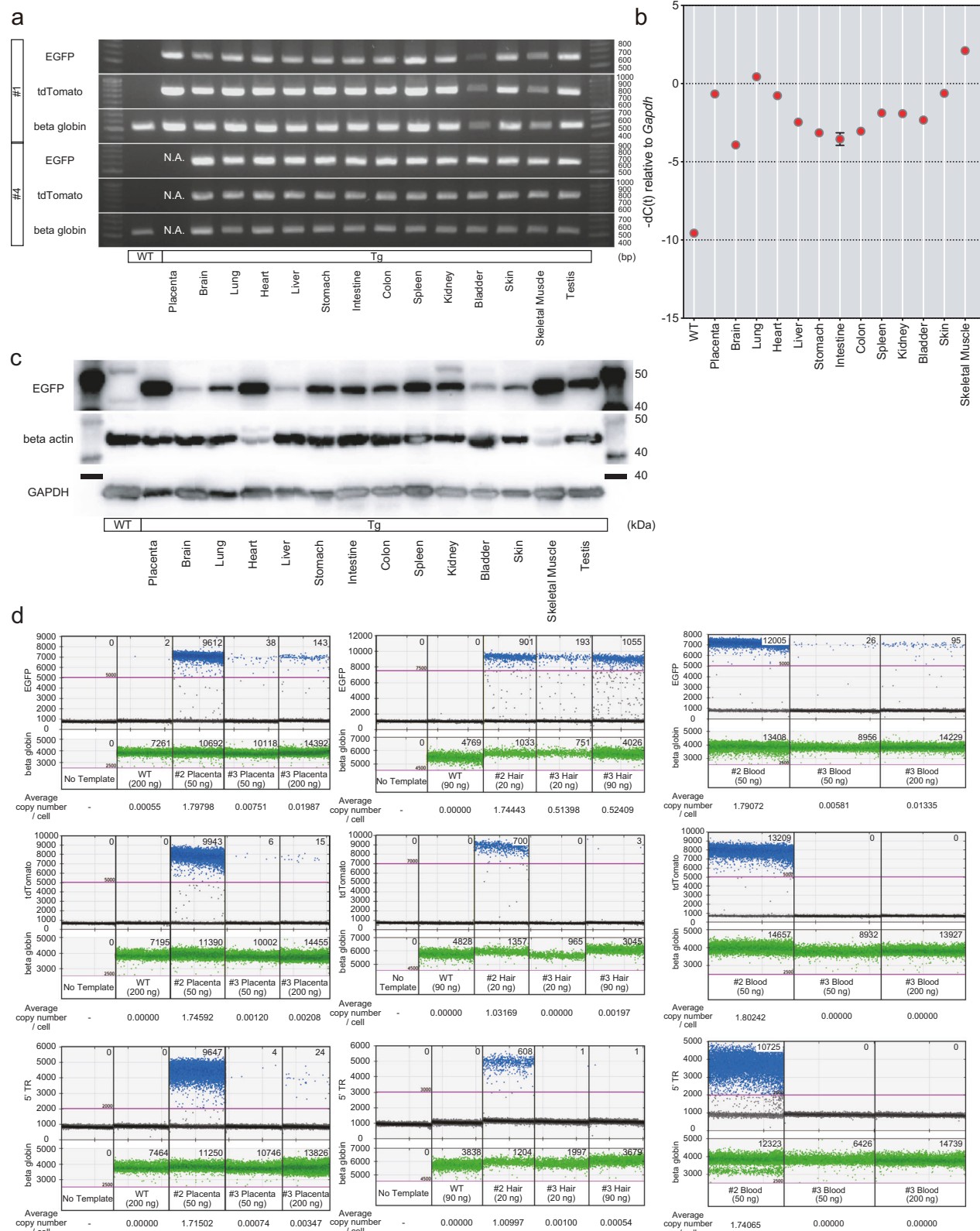

**Fig. 5 | Detection of the transgenes in the tissues of transgenic monkeys. a** The genomic PCR for the fluorescence reporters in the tissues of transgenic monkeys. **b** Expression of the EGFP in the tissues of transgenic monkey #1. Error bars, mean values + s.d. *n* = 3 technical replicates. **c** The Western blotting for the EGFP in the tissues of transgenic monkey #1. **d** The ddPCR for the fluorescence reporters and 5′ TR in the tissues of transgenic monkeys. Source data are provided as a Source Data file.

detected in the blood of monkey #3, indicating that the body of monkey #3 lacks piggyBac transgenes (Fig. 5d). On the other hand, EGFP signals were detected, albeit at markedly lower levels compared to monkey #2. However, it should be noted that EGFP signals were not completely absent even in wild-type (WT) samples with 200 ng of DNA in the ddPCR assay. This observation suggests that the EGFP primer used in ddPCR might have been more sensitive than other primers, potentially amplifying minimal contamination in the extracted

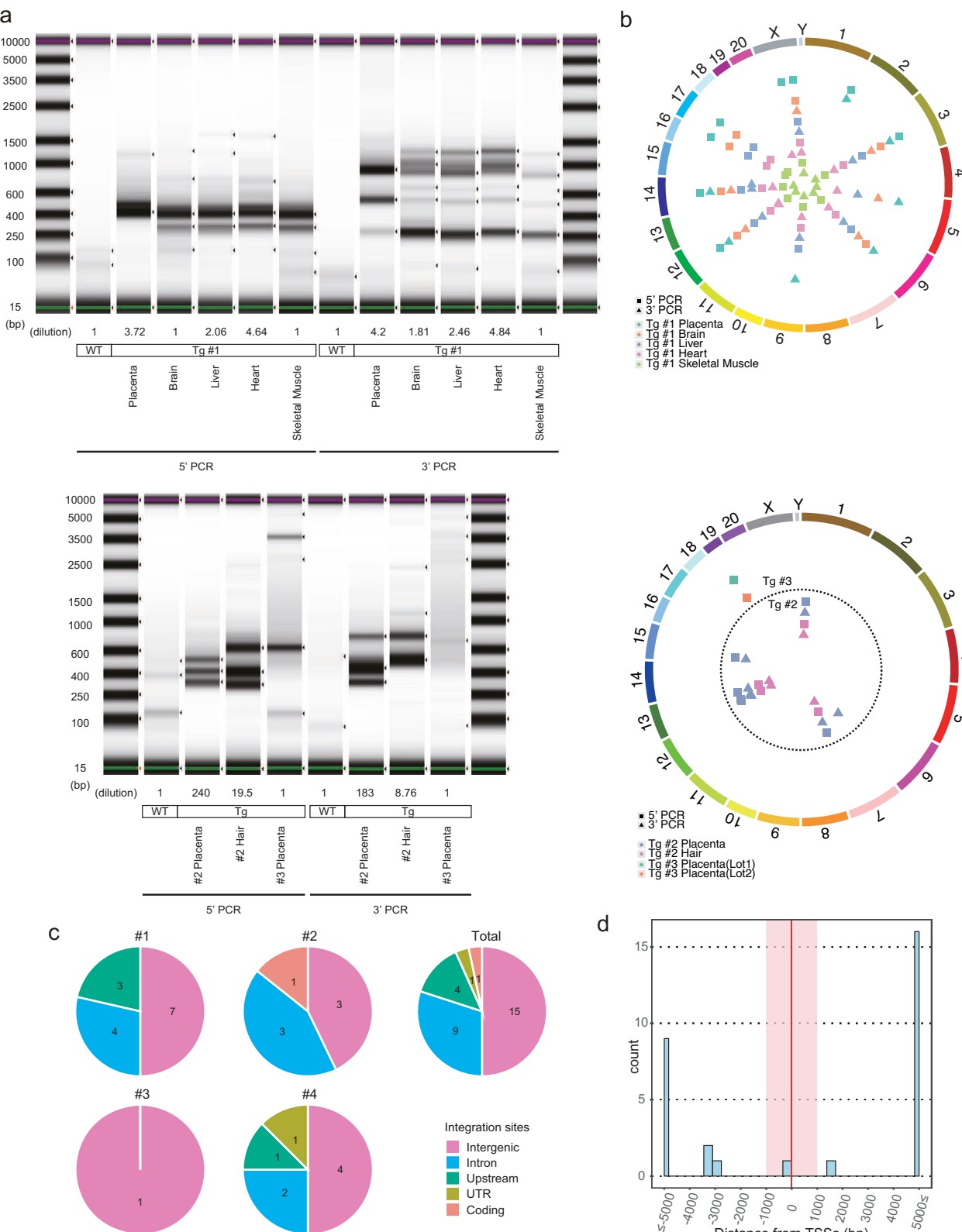

**Fig. 6 | Identification of the transgene insertion sites in the tissues of transgenic monkeys. a** Inverse PCR for detection of transgene insertions in each tissue. **b** NGS sequencing for mapping of the insertion sites. **c** Genomic region of the insertion sites. Upstream indicates the sites within 5000 bp of the transcription start sites. **d** Distance from the transcription start sites (TSSs). Source data are provided as a Source Data file.

genomic DNA or homologous sequences. These findings highlight the importance of validating results from multiple perspectives.

To identify the transgene insertion sites in these monkeys and evaluate any potential biases between individuals or among different tissues, we performed inverse PCR assays. The assays for the deceased monkeys (#1 and #4) showed that all tissues, except for the placenta, exhibited a similar band pattern, indicating that they have analogous insertion sites (Fig. 6a and Supplementary Fig. 8a). The assays for the

live monkey (#2) also revealed that the placenta and the hair showed different band patterns (Fig. 6a). Insertion sites were determined by next-generation sequencing (NGS) analysis, which revealed that most of the tissues from monkeys #1 and #4 harbored 9 and 8 common insertion sites, respectively (Fig. 6b and Supplementary Fig. 8b). The placenta and hair of monkey #2 were found to share 4 insertion sites (Fig. 6b). A unique insertion site was identified in the placenta of monkey #3 (Fig. 6b). In addition, the placentas of monkeys #1 and #2 exhibited additional specific insertion sites (Fig. 6b). A total of 30 integration sites were identified, with the following distribution: 15 in intergenic regions, 9 in introns, 4 upstream of genes, 1 in a UTR, and 1 in a coding region (Fig. 6c). This general pattern was consistent across the animals, including the two that did not survive (monkeys #1 and #4), and previous piggyBac reports[18,23]. Interestingly, we observed upstream integrations in both monkey #1 (MRPL48, WDR19) and monkey #4 (TLCD5), but their specific effects on development remain unclear, and further investigations are required to determine their potential roles.

Regarding the integration tendencies of piggyBac, we noted a difference from the observations of Huang et al., who reported a high preference for insertion within 5 kb of the transcription start sites (TSSs)[24]. In contrast, most of the integration sites in our monkeys were located outside this range (Fig. 6d). This raises the possibility that successful term development might be linked to an avoidance of transgene integration into critical genomic loci.

## Immunofluorescence analysis for the expression of transgenes

While RT-PCR and Western blotting analysis confirmed transgene expression in all examined tissues (Fig. 5b, c), fluorescence imaging at low magnification did not consistently reveal strong signals in some organs, such as the liver, spleen, lung and kidneys (Supplementary Figs. 5 and 6). Additionally, PCR analysis cannot exclude the possibility that the detected transgenes were derived from immune cells, which are present in all tissues. Therefore, to analyze the expression of transgenes at the single-cell level and distinguish tissue-specific expression from that of immune cells, we performed immunostaining and high-magnification imaging of tissue samples from monkey #1. This analysis confirmed the transgene expression in all examined tissues and demonstrated transgene expression in tissue-specific cells that were morphologically distinct from immune cells (Fig. 7a–d and Supplementary Fig. 9).

Quantitative analysis of the immunofluorescence staining revealed that the brain and the liver showed relatively low expression rates and intensities of GFP, whereas the heart and the skeletal muscle showed nearly complete (95%–100%) positive rates and most of the cells showed high expression intensities (Fig. 7e, f), which was consistent with the results of RT-PCR and Western blotting (Fig. 5b, c). The proportion of fluorescent-positive cells in the liver, spleen, lung, and kidneys, which exhibited no clear fluorescent signal at low magnification, was relatively low, and the fluorescence intensity of these cells was also weak (Fig. 7b and Supplementary Fig. 9). This limited cell positivity, combined with low fluorescence intensity, likely explains the weak or undetectable fluorescence observed in the initial low-magnification images. In addition, these experiments revealed that the transgene expression showed variation among cells, i.e., mosaicism (Fig. 7a–d, f and Supplementary Fig. 9a–c and e).

To elucidate the cause of the mosaicism, fibroblasts were cultured from the skin of transgenic monkey #1 and cells with strong-positive, weak-positive, and negative transgene expression were collected (Fig. 8a–c). Genotyping analysis of these cells revealed that all three populations, including the population that was negative for transgene expression, had transgene insertion (Fig. 8d). In the inverse PCR assay, the negative, weak-positive, and strong-positive fibroblasts exhibited different band patterns (Fig. 8e). In addition, the band pattern for the brain appeared to be an assembly of the band patterns of the three

fibroblast populations. Furthermore, NGS analysis revealed that the insertion site patterns differed among the negative, weak-positive, and strong-positive fibroblasts, indicating genetic mosaicism within the tissue (Fig. 8f).

To explore the relationship between transgene mosaicism and DNA methylation, we conducted bisulfite PCR and analyzed the methylation status at the CAG promoter of the transgene in fibroblast subpopulations with strongly positive and negative fluorescence[25]. Unexpectedly, no significant differences in methylation levels were observed between these fractions (Fig. 8g). We then further analyzed tissues with varying levels of transgene expression, including the brain and liver (low expression) and the heart and skeletal muscle (high expression). While a trend toward hypomethylation was observed in the heart, no statistically significant differences were found (Fig. 8h). These findings suggest that transgene mosaicism cannot be fully explained by DNA methylation alone and may involve other epigenetic regulatory mechanisms, such as histone modifications or chromatin structure alterations.

## Germline integration of transgenes in the monkey

Finally, to examine the germ-cell contribution of transgenes in the transgenic monkey, the testes of transgenic monkey #1 were stained with anti-DDX4 (MVH) antibody (Fig. 9a). Interestingly, quantitative analysis of the immunofluorescence staining revealed that the GFP-positive rate and intensity of the DDX4-negative cells, the major population in testes, in the transgenic testes were lower than those of the DDX4-positive germ cells (Fig. 9b, c). These results are consistent with the relatively weak fluorescence in the whole testes at low magnification (Supplementary Fig. 5). Importantly, more than 80% of the DDX4-positive germ cells expressed GFP (Fig. 9b), demonstrating transgene integration in the germ cells.

## Discussion

Before the present study, it was unclear whether non-viral methods could be used to generate transgenic animals in non-human primates[18]. In the present study, we optimized the conditions for the production of transgenic cynomolgus monkeys and succeeded in generating transgenic monkeys by non-viral piggyBac transposition.

During these procedures, we attempted to determine the sex of the embryos and succeeded in male detection by genotyping tro-phectoderm cells using the laser-assisted hatching method (Fig. 4d). Although the implantation rate of genotyped embryos (1/9, 11.1%) tended to be lower than that of non-genotyped embryos (3/11, 27.3%), we speculate that this method would also be useful for preimplantation genotyping of other genetically modified embryos, such as those modified using CRISPR/Cas9 methods.

The identification of transgene insertion positions revealed that a maximum of only 4 insertion sites were detected in the surviving monkeys in this study, compared to 8 to 9 in the monkey that died after birth and the monkey that was stillborn. This result raises the possibility that the number of transgene insertions affected survival, although the small sample size makes it difficult to draw definitive conclusions. In a previous study on gene transfer in rhesus monkey embryos using piggyBac, it was reported that embryos with arrested development had more transgene insertions than those that developed to the blastocyst stage[18]. The results showed a correlation between the number of transgenes and the level of developmental toxicity, which was consistent with our results.

The rates and intensities of transgene expression in the tissues of transgenic monkey #1 varied among and within tissues (Figs. 5b and 7). The expression of transgenes in the brain and in the endoderm-derived tissues, including the liver, stomach, intestine, and the colon tissues, was lower than the expression in the other tissues. However, the expression in the mesoderm-derived tissues, including the heart and the skeletal muscle tissues, was relatively high. This expression

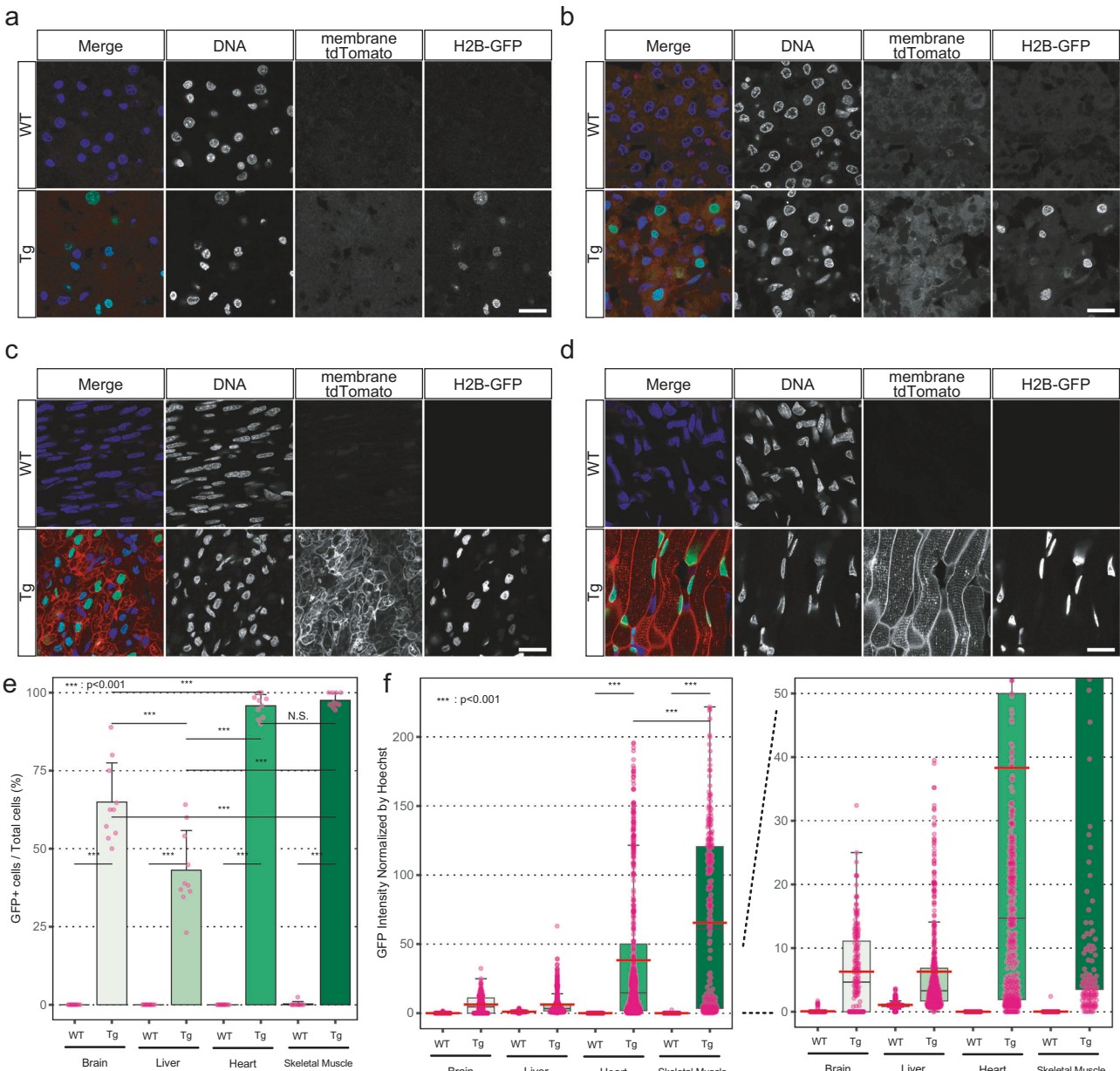

**Fig. 7 | Immunofluorescence analysis for the expression of transgenes. a–d**
Expression of fluorescence reporters in the brain, liver, heart, and skeletal muscle, respectively. Scale bar, 20 μm. **e** Bar graph showing the GFP-positive cell rates based on the analysis of 10 fields per sample. Error bars, mean values + s.d.; $n = 10$ fields in each sample. One-way ANOVA and Tukey-Kramer post-hoc contrasts were used for the comparisons. ***$P < 0.001$; N.S. not significant. **f** Box plots of GFP intensities in each nucleus of the images. The top and bottom edges of boxes indicate the first and third quartiles, respectively; the center lines indicate the medians; the ends of whiskers indicate the maximum and minimum values within 1.5 times the inter-quartile range; and the red lines indicate the means. $n = 10$ fields in each sample. One-way ANOVA and Tukey-Kramer post-hoc contrasts were used for the comparisons. ***$P < 0.001$. Source data and $P$ values are provided as a Source Data file.

variation among tissues might have been caused by the CAG promoter. The CAG promoter consists of the human cytomegalovirus (HCMV) immediate-early enhancer, the promoter, the first exon and the first intron of the chicken beta-actin, and the second intron and third exon of the rabbit hemoglobin subunit beta-1/2. The HCMV enhancer is known to show strong activity in ectoderm-derived tissues, and the beta-actin promoter is known to show strong activity in mesoderm-derived tissues[26,27]. Therefore, the CAG promoter does not have an expression control region that shows strong activity in endoderm-derived tissues[28]. Our results are consistent with the properties of the CAG promoter. Notably, similar results have been reported for other transgenic animals generated using the CAG promoter, including mice, rats, and rabbits[29–31].

In transgenic monkey #1, expression mosaicism was observed (Figs. 7 and 8). The mosaicism was caused, at least in part, by genetic mosaicism within the tissues (Fig. 8). Given that almost all cells in several tissues, including the heart, the skeletal muscle, and the DDX4-positive germ cells, expressed transgenes and that major cell populations of fibroblasts, including the population negative for transgene expression, had transgene insertions, most cells composing the transgenic monkey are likely to harbor the transgene insertions. Therefore, in part, position effect variegation and epigenetic regulation might contribute to transgene silencing.

Previous studies using lentiviral vectors have reported significant silencing attributed to global hypermethylation of the transgene promoter[32–34]. In contrast, our DNA methylation analysis

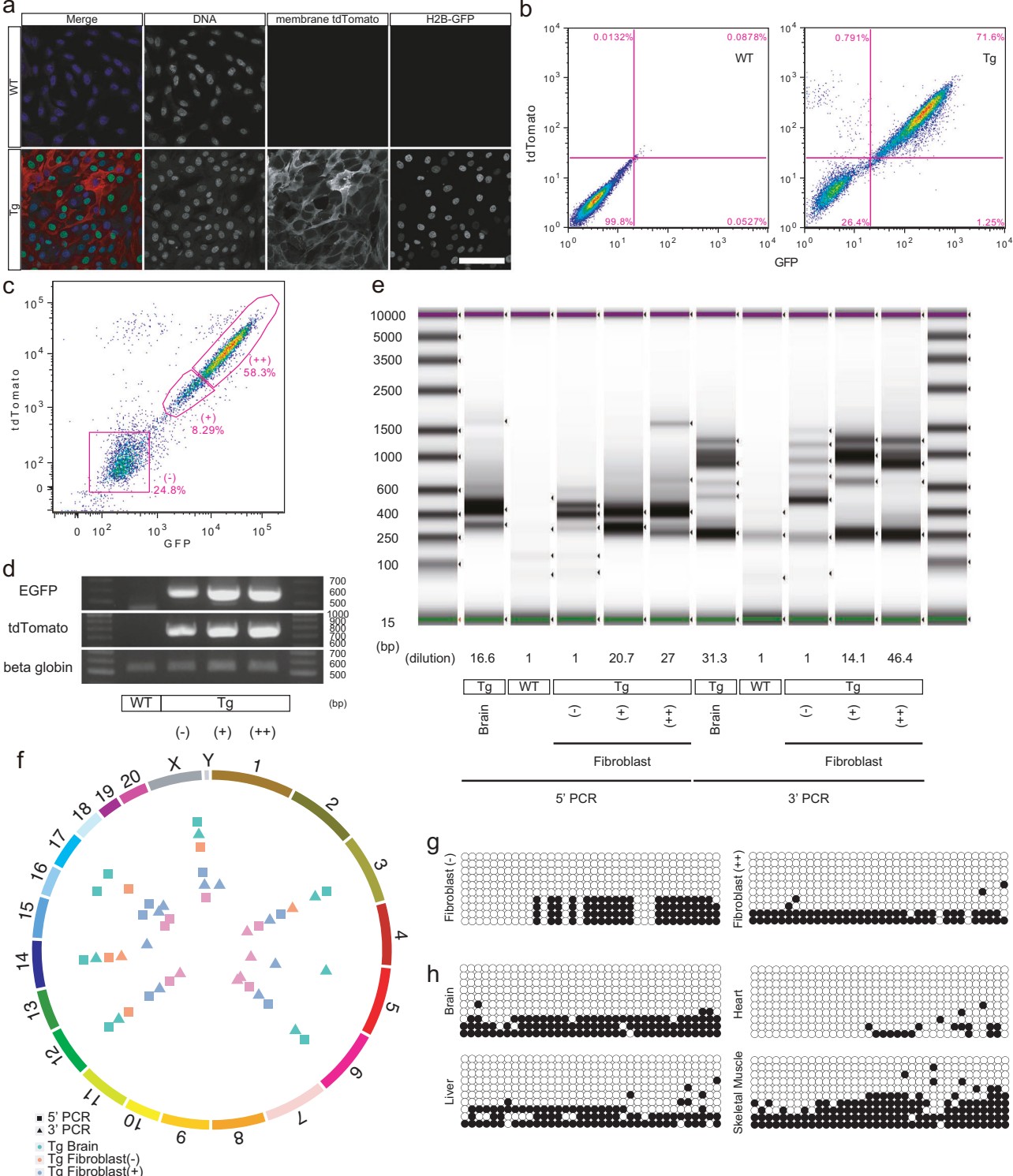

**Fig. 8 | Characterization of fluorescence-positive and -negative fibroblasts from the transgenic monkey. a** Expressions of the fluorescence reporters in the fibroblasts from the transgenic monkey. Scale bar, 100 µm. **b** FACS plot of the expression of fluorescence reporters in the fibroblasts. **c** Sorting of fibroblast populations that showed negative, weak-positive, and strong-positive transgene expression. **d** The genomic PCR for the fluorescence reporters in the fibroblast populations. **e** Inverse PCR for detection of transgene insertions in the fibroblast populations.

For a clear comparison of band patterns, the PCR amplicons of the fibroblast populations that showed weak- and strong-positive transgene expression, and that of the brain, were diluted. **f** NGS sequencing for mapping the insertion sites. **g** Methylation levels of CAG promoter in the fibroblasts. White and black circles indicate unmethylated and methylated cytosines, respectively. **h** Methylation levels of CAG promoter in the brain, liver, heart, and skeletal muscle. Source data are provided as a Source Data file.

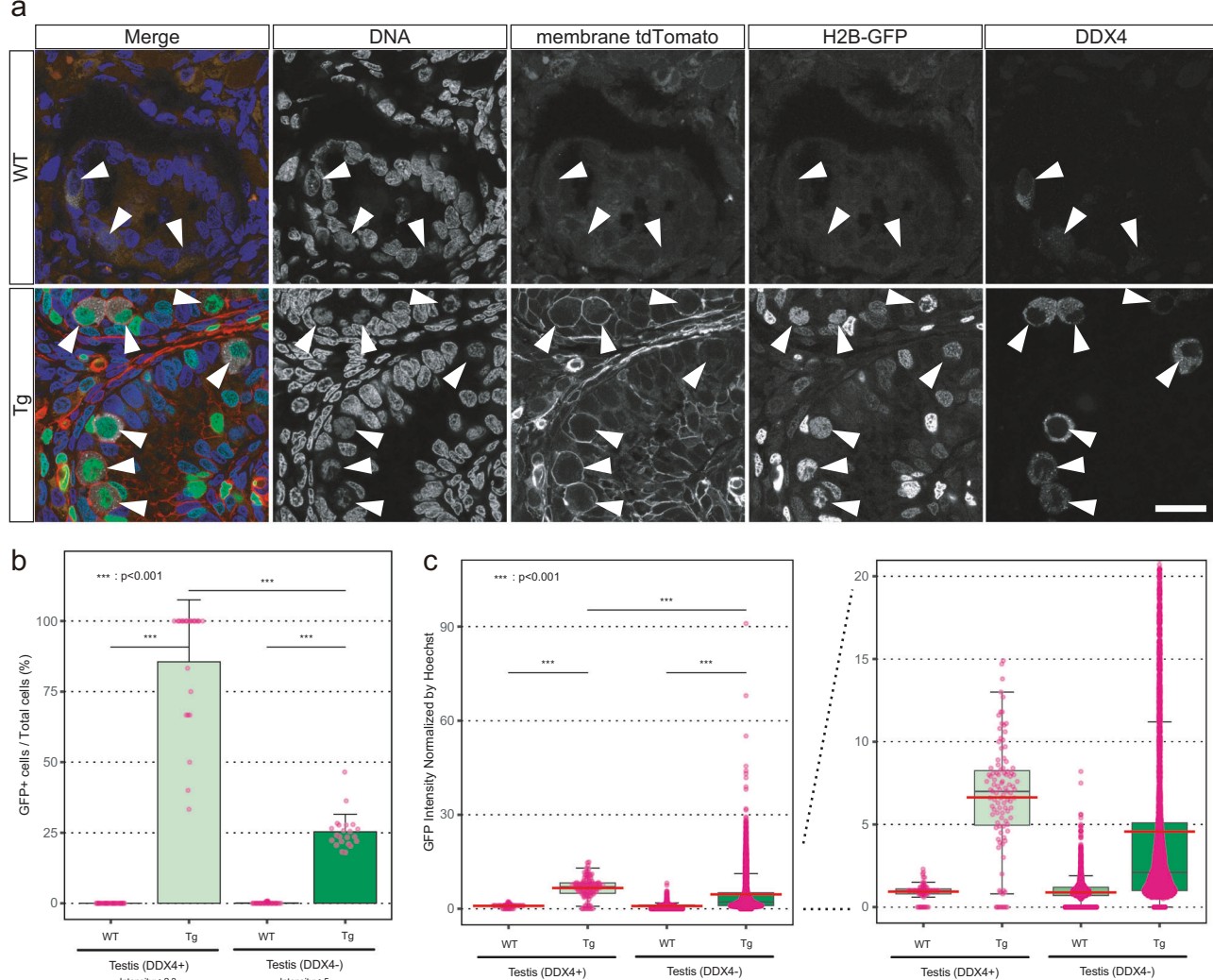

**Fig. 9 | Germline transmission of transgenes in the monkey. a** Expression of fluorescence reporters and a germ-cell marker, DDX4, in the testes. Scale bar, 20 μm. **b** Bar graph of GFP-positive rates in each image. Error bars, mean values + s.d.; $n = 27$ fields in the wildtype sample; $n = 22$ fields in the transgenic sample. One-way ANOVA and Tukey-Kramer post-hoc contrasts were used for the comparisons. ***$P < 0.001$. **c** Box plots of GFP intensities in each nucleus of the images. The top and bottom edges of boxes indicate the first and third quartiles, respectively; the center lines indicate the medians; the ends of whiskers indicate the maximum and minimum values within 1.5 times the interquartile range; and the red lines indicate the means. $n = 27$ fields in the wildtype sample; $n = 22$ fields in the transgenic sample. One-way ANOVA and Tukey-Kramer post-hoc contrasts were used for the comparisons. ***$P < 0.001$. Source data and $P$ values are provided as a Source Data file.

of the CAG promoter in the piggyBac transgenic monkey demonstrated mixes of hypomethylated and hypermethylated sequences within the same tissues (Fig. 8g, h). This suggests that position effects, rather than piggyBac-specific methylation, are the dominant factor influencing DNA methylation. Additionally, our germline transmission analysis in mice revealed robust fluorescence in a substantial proportion of F2 embryos (Fig. 2f). Although we acknowledge that the transgene may behave differently in mice and non-human primates, the finding is in stark contrast to a previous study using a lentiviral vector, which reported increased DNA methylation leading to silencing over generations[34]. These findings align with previous reports that highlighted piggyBac's relative resistance to epigenetic silencing compared to lentiviral transgenes[23,35].

The genetic mosaicism of transgenes indicates that transgene insertions into the genome occurred after the first DNA replication in the one-cell stage embryo. Interestingly, mosaicism occurs in all cases of genetically modified embryos with lentiviral vectors[10,11]. In the case of fertilized eggs, uniform gene transfer into all cells is possible only

when the gene transfer is completed before DNA replication during the one-cell stage. However, because proviral DNA is synthesized by reverse transcription in host cells before genome insertion, its insertion into the genome requires a certain amount of time ($>48$ h)[17]. These problems are considered difficult to overcome in principle. By contrast, because transgene insertion into a genome in the piggyBac system depends on the activity of the PBase, the transgene insertion period can be limited by controlling the PBase activity. This capability introduces the possibility of eliminating mosaicism in transgenic animals in the future.

However, because the PBase used in this study is not actively degraded and was not designed with an artificial regulatory system, the activity of the PBase remained after the first DNA replication, resulting in genetic mosaicism in the transgenic embryo. Unexpectedly, the placenta of the transgenic monkeys showed patterns of transgene insertion that differed from that of the other tissues (Fig. 6). These results indicate that the PBase activity might remain until the first differentiation into trophectoderm. This phenomenon is inconvenient for genotyping using the placenta of delivered monkeys. Therefore, we

believe that artificial control of the PBase activity will be necessary to completely prevent mosaicism[36].

The collective results show that we succeeded in generating transgenic non-human primates by a non-viral method using piggyBac transposition optimized for the production of transgenic cynomolgus monkeys. Both piggyBac and lentiviral vectors share the limitation of random integration, which can lead to insertional mutagenesis or embryotoxicity due to excessive genomic insertions. Furthermore, the relatively low transgene expression observed in the brain in our study suggests that the choice of promoter may significantly affect expression levels across different tissues. These issues highlight the need for advancements in targeted integration techniques. However, the current efficiency of knock-in methods, particularly for large transgene fragments, remains suboptimal[7,8]. Despite this shared limitation, piggyBac offers a number of advantages. As a non-viral system, it reduces biosafety concerns and eliminates the need for complex procedures, such as the production of viral particles and titration of infection activity. This makes it a simpler and more accessible approach for generating transgenic animals in large species, including non-human primates. Additionally, in the case of lentiviral transgenic monkey generation, lentiviral-production-derived debris can exhibit fluorescence and hinder the evaluation of transgene expression[10]. By contrast, our method was able to precisely evaluate the fluorescence before embryo transfer (Fig. 4c–g). We believe that our method will provide a practical non-viral protocol for the generation of transgenic non-human primates.

## Methods

### Ethical statement
We followed the Reporting in Vivo Experiments (ARRIVE) guidelines developed by the National Centre for the Replacement, Refinement & Reduction of Animals in Research (NC3Rs). We also followed "The Act on Welfare and Management of Animals" from the Ministry of the Environment, the "Fundamental Guidelines for Proper Conduct of Animal Experiment and Related Activities in Academic Research Institutions" under the jurisdiction of the Ministry of Education, Culture, Sports, Science and Technology, and the "Guidelines for the Proper Conduct of Animal Experiments" from the Science Council of Japan. All animal experimental procedures were approved by the Animal Care and Use Committee of Shiga University of Medical Science (approval numbers: 2019-7-12, 2021-8-3, 2022-6-13 and 2024-1-3).

### Vector construction
To construct the PB-CAG-membrane tdTomato-2A-H2B-GFP, an amplified PCR product from pCAG-TAG was cloned into pPB-CAG-cHA-pA. pCAG-TAG was purchased from Addgene (Plasmid #26771)[37]. To generate pcDNA3.1-hyPBase-poly(A83) containing piggyBac transposase, a hyPBase insert from pCMV-hyPBase was introduced into pcDNA3.1-H2B-mCherry-poly(A83)[38]. The primer sequences are shown in Supplementary Table 1.

### In vitro transcription of PBase mRNA
Linearized pcDNA3.1-hyPBase-poly(A83) was treated with 0.5% sodium dodecyl sulfate (SDS) and 0.2 mg/mL proteinase K for 30 min at 50 °C, purified with phenol–chloroform, and precipitated with ethanol. The purified DNA was used as the template for in vitro transcription. The PBase mRNA was transcribed using an mMESSAGE mMACHINE T7 Transcription Kit (Thermo Fisher Scientific, AM1345). The mRNA was purified with a MEGAclear Transcription Clean-Up Kit (Thermo Fisher Scientific, AM1354).

### Oocyte and sperm collection in mice
Eight- to eighteen-week-old B6D2F1 female mice (Japan SLC) were superovulated by injection of 7.5 IU of pregnant mare serum gonadotropin (PMSG) (Serotropin; ASUKA Pharmaceutical), followed by injection of 7.5 IU of human chorionic gonadotropin (hCG) (Gonatropin; ASUKA Pharmaceutical) 48 h later. Cumulus-oocyte complexes (COCs) containing unfertilized eggs were harvested 16 h after hCG injection and placed in human tubal fluid (HTF) medium (ARK Resource, I0BAIH200).

Spermatozoa were collected from the cauda epididymis of 11- to 15-week-old B6D2F1 male mice (Japan SLC) and cultured for 1.5 h in HTF medium.

### IVF and PN injection in mice
For IVF, the spermatozoa were introduced into fertilization droplets containing COCs at a final concentration of $1 \times 10^6$ cells/mL after pre-incubation. After incubating for 6 h, the fertilized embryos were collected and washed three times in potassium simplex optimized medium (KSOM) (ARK Resource, I0BAIK200).

For PN injection, approximately 5–10 pL of 50 ng/µL hyPBase mRNA and various concentrations of PB-CAG-membrane tdTomato-2A-H2B-GFP in Opti-MEM (Thermo Fisher Scientific, 22600-050) were injected into the pronucleus of 1-cell embryos 6 h after insemination using a micromanipulator (Narishige). After injection, the embryos were cultured in KSOM medium under mineral oil (Sigma, M8410).

### ICSI and co-injection in mice
Mouse ICSI was performed according to the method described previously[39]. Briefly, oocytes were harvested as previously described. The sperm heads were injected into metaphase II (MII)-stage oocytes in M2 medium (Sigma, M7167) using a micromanipulator and a piezo impact-driving unit (Prime Tech). For co-injection, the piggyBac vector and the PBase mRNA in Opti-MEM were co-injected with sperm during ICSI[8,40]. The ICSI embryos were cultured in microdrops of KSOM medium until the embryos developed to the blastocyst stage.

### Generation of transgenic mice
Fluorescence-positive blastocyst embryos were selected and subsequently transferred into the uterus of pseudopregnant recipient mice. The offspring were recovered at E19.5. The presence of transgenes in the offspring was examined by fluorescence, and the presence of the enhanced green fluorescent protein (EGFP) gene and tdTomato gene were examined by genomic PCR. The primer sequences are shown in Supplementary Table 1.

### Embryo imaging
Blastocyst embryos were fixed in 4% PFA/PBS at room temperature for 15 min. After being washed three times with 0.1% Tween-20/PBS, the samples were permeabilized with 0.5% Triton X-100/PBS at room temperature for 30 min and then washed once with 3 mg/mL PVP/PBS and twice with 0.1% Tween-20/PBS. The samples were incubated with Hoechst 33342 overnight at 4 °C. After the samples were washed three times with 0.1% Tween-20/PBS, the tdTomato, EGFP, and Hoechst signals were observed using a confocal laser scanning microscope (SP8; Leica Microsystems).

### Quantitative analysis of confocal images
Photographs were taken at the same exposure settings within the same experiments. Image analysis was performed using Imaris (Bitplane) and Fiji running ImageJ software (US National Institutes of Health). The procedure used to quantify fluorescence signals using the software consisted of manually selecting each nucleus and extracting the median intensities. Fluorescence intensities were normalized by the median nuclear fluorescence intensity for Hoechst. Cells with a normalized fluorescence intensity greater than 2 were defined as fluorescence-positive.

## Oocyte and sperm collection in cynomolgus monkeys

Female cynomolgus monkeys, aged between 4 and 13 years, were used for oocyte collection. The animals were housed under a 12-h light regimen (8:00 AM to 8:00 PM) and fed a daily diet consisting of 20 g of commercial monkey chow (CMK-1; CLEA Japan) per kilogram of body weight each morning, with an additional 20–50 g of sweet potato provided in the afternoon. Water was available *ad libitum*, and the animal rooms were maintained at $25 \pm 2\,°C$ with a relative humidity of $50 \pm 5\%$. Two weeks after administering a subcutaneous injection of 0.9 mg of a gonadotropin-releasing hormone antagonist (Leuplin for Injection Kit; Takeda Chemical Industries), the monkeys were anesthetized and implanted subcutaneously with a micro-infusion pump (iPRECIO; ALZET Osmotic Pumps, SMP-200). This pump delivered human follicle-stimulating hormone (hFSH; uFSH, Asuka Pharmaceutical) at a dose of 15 IU/kg at a continuous rate of 7 μL/h over a period of 10 days[10,22]. Following this treatment, an intramuscular injection of 400 IU/kg human chorionic gonadotropin (hCG; Gonatropin, Asuka Pharmaceutical) was administered. Forty hours after the hCG injection, oocytes were retrieved via follicular aspiration performed with laparoscopic guidance (Machida Endoscope, LA-6500). Fresh sperm were collected by applying electrical stimulation to the penis without the use of anesthesia.

## ICSI and co-injection in cynomolgus monkeys

COCs were initially collected in Alpha Modification of Eagle's Medium (MP Biomedicals, 09103112-CF) supplemented with 10% serum substitute (Irvine Scientific, 99193). Cumulus cells were then removed by incubating the COCs in 0.5 mg/mL hyaluronidase (Sigma, H4272). ICSI was subsequently performed on MII-stage oocytes suspended in mTALP medium containing HEPES, using a micromanipulation system. During the ICSI procedure, sperm were co-injected along with the piggyBac vector and PBase mRNA[8,40]. Prior to injection, both the MII oocytes and sperm were rinsed in a droplet of Opti-MEM containing the piggyBac vector and PBase mRNA, and the injection was carried out within this droplet.

## Preimplantation genotyping with laser-assisted hatching

Following ICSI, embryos were cultured in CMRL Medium-1066 (Thermo Fisher Scientific, 21540026) supplemented with 20% bovine serum (Thermo Fisher Scientific) at $38\,°C$ in 5% $CO_2$ and 5% $O_2$. The zona pellucida was drilled with a laser under microscopy with a micromanipulator and Saturn 5 laser system (CooperSurgical) at the four-cell stage. The hatched trophectoderm cells were then sampled with a micromanipulator and Saturn 5 laser system. After genomic DNAs were extracted from the samples, genomic PCR was used to examine for the presence of a Y-chromosome-specific gene. The primer sequences are shown in Supplementary Table 1.

## Generation of transgenic cynomolgus monkeys

Photographs were taken at the same exposure settings within the same experiments. Quantitative analysis of tdTomato expression was performed using Fiji running the ImageJ software. After tdTomato-positive embryos were selected, one embryo was transferred into each appropriate recipient female[10,22]. The menstrual cycles of all female monkeys were monitored daily, and potential recipients were identified based on their reproductive cycles. The final selection of surrogate mothers was determined through laparoscopic examination of follicular development and ovulation scars to ensure the best possible synchronization for embryo implantation. Embryos were aspirated into a catheter (Kitazato Medical Service, ETC3040SM5-17) under a stereomicroscope. The catheter was inserted into the oviduct of the recipient via the fimbria under a laparoscope, and the cultured embryo was transplanted with a small amount of medium. Pregnancy was determined by ultrasonography 30 days after ICSI. The presence of the transgenes in the offspring was examined by genomic PCR and droplet digital PCR (ddPCR; BioRad). The primer and probe sequences are shown in Supplementary Table 1.

## Quantitative RT-PCR

Total RNAs were extracted from the tissues of the transgenic monkey using RNeasy Mini kits (Qiagen, 74104). For reverse transcription, ReverTra Ace (Toyobo, TRT-101) and oligo (dT)20 primer were used. For real-time PCR, THUNDERBIRD SYBR qPCR Mix (Toyobo, QPS-201) was used. Transcript levels were determined in triplicate reactions and normalized against the corresponding levels of Gapdh. The primer sequences are shown in Supplementary Table 1.

## Western blotting

For Western blotting, tissues were fixed by 10% trichloroacetic acid immediately after freeze fracturing to avoid protein degradation. The fixed samples were treated with SDS sample buffer (Wako, 198-13282) to solubilize proteins. Following sonication, the protein concentration in each sample was quantified using XL-Bradford (SDS-PAGE compatible; APRO SCIENCE, KY-1030). Equal amounts of protein (10 μg per sample) were then separated by SDS-PAGE. After blotting, the membranes were blocked with 5% skim milk/0.1% Tween-20/TBS for 1 h at room temperature and incubated with anti-GFP antibodies (1:2000 dilution; Thermo Fisher Scientific, A21311) in 1% skim milk/0.1% Tween-20/TBS overnight at $4\,°C$. After washing three times with 0.1% Tween-20/TBS, the membranes were incubated with HRP-conjugated secondary antibodies (1:2000 dilution; Thermo Fisher Scientific, 65-6120) in 1% skim milk/0.1% Tween-20/TBS for 1 h at room temperature. After washing three times with 0.1% Tween-20/TBS, immunoreactive proteins were detected with enhanced chemiluminescence (Cemi-Lumi One Super; Nacalai, 02230-30) and a FUSION imager (VILBER). The membranes were stripped with WB Stripping Solution Strong (Nacalai, 05677-65), and the beta-actin proteins and gapdh proteins were detected with Anti-beta-actin pAb-HRP-DirectT (1:2000 dilution; MBL, PM053-7) and Anti-GAPDH pAb-HRP-DirectT (1:8000 dilution; MBL, M171-7). For total protein detection, the membranes were stained with Ponceau S staining solution (APRO SCIENCE, SP-4030).

## Identification of piggyBac insertion sites

The genomic DNA extracted from tissue samples of 500 ng at most was first subjected to restriction digestion with 30 U of Dpn II (NEB, R0543) in a 100 μL reaction volume. Subsequently, 10 U of T4 DNA ligase (Thermo Fisher Scientific, EL0011) was added along with T4 DNA ligation buffer to a total reaction volume of 400 μL. After the samples were incubated at $4\,°C$ overnight, DNA was purified and eluted in 30 μL elution buffer (10 mM Tris-HCl pH 8.5) with NucleoSpin Gel and PCR Clean-up (Takara, 740609). Using the purified DNA as templates, we performed inverse PCR with the primers listed in Supplementary Table 1 to amplify self-ligated DNA fragments containing each of the 5' and 3' ends of the piggyBac transposon inserted in the genome. After the amplicon was purified with 1.2× volumes of AMPure XP beads (Beckman Coulter, A63881), approximately one-fourth of the eluted DNA was subjected to a second round of PCR with a combination of the forward and reverse primers listed in Supplementary Table 1. The amounts of DNA used for the first PCR ranged from 1 to 5 ng for the samples shown in Fig. 6 and Supplementary Fig. 8. The DNA amount was 25 ng for the samples in Fig. 8. The PCR cycle numbers for the first and second PCR were 25 and 7, respectively, for the samples in Fig. 6 and Supplementary Fig. 8, and they were 23 and 12, respectively, for the samples in Fig. 8. The resultant DNA libraries were purified with 1.2× AMPure XP beads (Beckman Coulter, A63881). To analyze the size distributions, we loaded at most 4 ng of the library after dilution (when necessary) into a 4200 TapeStation system (Agilent) using a High Sensitivity D5000 ScreenTape and Reagent Kit (Agilent, 5067–5593). After pooling the libraries, we sequenced them using an iSeq 100 system (Illumina) and iSeq 100 i1 reagent (300 cycles) (Illumina, 20021534).

On the basis of the index sequences, raw sequenced reads were demultiplexed using bcl2fastq (v.2.20; Illumina). Only the reads having the assumed piggyBac-mediated insertion sequences and Dpn II

recognition sites were retained for further analyses. After trimming adaptor sequences and piggyBac-mediated insertion sequences with Cutadapt (v.3.5), we mapped the trimmed reads to the autosomal chromosomes (from chr1 to chr20) of the MacFas5 genome (the cynomolgus macaque (*Macaca fascicularis*)) and the sex chromosomes (chrX and chrY) of the Mmul_8.0.1 (rheMac8) genome (the Rhesus monkey (*Macaca mulatta*)) using Bowtie 2 (v.2.4.5) with the --maxins 5000 option[41,42]. The uniquely and properly mapped reads were summarized and counted for each insertion site (Source Data file). The nearest genes to each insertion site were identified on the basis of the autosomal chromosomes of Macaca_fascicularis.Macaca_fascicularis_5.0.102.chr.gtf and the sex chromosomes of Macaca_mulatta.Mmul_8.0.1.97.chr.gtf file obtained from the Ensembl genome database. The insertion sites were visualized on Circos plots (Figs. 6, 8, and Supplementary Fig. 8) using the R (v.4.0.3) package circlize (v.0.4.10)[43]. To avoid misannotation due to various biases, only insertion sites supported by more than 10, 100, and 10 sequenced reads in Figs. 6, 8, and Supplementary Fig. 8, respectively, were conservatively selected as the piggyBac insertion sites.

### Tissue sectioning and immunohistochemistry

For tissue sectioning, the tissues were fixed in 4% paraformaldehyde solution overnight at 4 °C. The samples were then immersed successively in 10%, 20%, and 30% sucrose in PBS, embedded in optimal cutting temperature (OCT) compound embedding medium (Leica Microsystems, 8097753), and frozen. The frozen blocks were sectioned at a thickness of 10 μm on a cryostat (Leica Microsystems) and mounted onto glass slides (Platinum Pro; Matsunami, PRO-01). Air-dried sections were washed with PBS and permeabilized with 0.2% Triton X-100/PBS at room temperature for 30 min. The slides were then blocked with 10% normal donkey serum/2% skim milk/0.1% Tween-20/PBS for 1 h at 4 °C and incubated with primary antibodies in the blocking solution overnight at 4 °C. After the slides were washed three times with 0.1% Tween-20/PBS, they were incubated with tetramethylrhodamine (TRITC) or Alexa Fluor 647-conjugated secondary antibodies and Hoechst 33342 in 2% skim milk/0.1% Tween-20/PBS for 4 h at 4 °C. After the slides were washed three times with 0.1% Tween-20/PBS, they were mounted with mounting medium (Vectashield; Vector Laboratories, VEC-H-1900-10). The antibodies used in the present study were anti-GFP, Alexa Fluor 488-conjugated (1:300 dilution; Thermo Fisher Scientific, A21311), anti-RFP (5F8) (1:1000 dilution; ChromoTek, PGI-5F8), and anti-DDX4 (1:100 dilution; Abcam, ab27591). Quantitative analysis of GFP expression was performed in 10 randomly selected fields per sample of brain, liver, heart, skeletal muscle, spleen, lung, and kidneys. In the analysis of germline integration, 27 and 22 randomly selected fields per sample of wildtype testes and transgenic testes, respectively, were used.

### Bisulfite PCR and DNA sequencing

The genomic DNA extracted from tissue samples was treated with an EpiTect Bisulfite Kit (QIAGEN, 59104) according to the manufacturer's instructions. The bisulfite-treated DNA was amplified by PCR for the CAG promoter region containing 36 CpG sites[25]. For PCR, KOD -Multi & Epi- (Toyobo, KME-101) was used, and the protocol consisted of 94 °C for 2 min, followed by 40 three-step cycles of 98 °C for 10 s, 60 °C for 30 s and 68 °C for 30 s. For DNA sequencing, the PCR products were cloned into the EcoRV site of pBRBlue II. For each sample, multiple PCR products from different tubes were cloned to reduce bias. To determine the DNA methylation status, ten clones from each sample were sequenced and analyzed with QUMA[44]. The primer sequences are shown in Supplementary Table 1.

### Statistics and reproducibility

One-way ANOVA and Tukey-Kramer post-hoc contrasts were used for the comparisons in Figs. 1e–h, 3b, c, 7e, f, 9b, c, and Supplementary Fig. 9d, e. The minimum number of animals necessary to achieve the

scientific objectives was used because of the ethical reason. No data were excluded. Some experiments producing the key data were repeated by different co-authors. Multiple animals were used for each biological replicate. All data were checked by multiple individuals who didn't know the genotype of animals.

### Reporting summary

Further information on research design is available in the Nature Portfolio Reporting Summary linked to this article.

## Data availability

The source data underlying Figs. 1e–h, 2d, e, 3b, c, 4b, 5a–c, 6b–d, 7e, f, 8d, f–h, 9b, c and Supplementary Figs. 4, 8b, and 9d, e are provided as a Source Data file. The sequencing data generated in this study are available under DDBJ bioproject number PRJDB18586. Source data are provided with this paper.

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

## Acknowledgements

We are grateful to Dr. Spyros Goulas (Kyoto University) and Dr. Takashi Hiiragi (Kyoto University and the Hubrecht Institute) for invaluable comments and advice on this work. We thank the Research Center for Animal Life Science and the Central Research Laboratory in Shiga University of Medical Science, and the Single-Cell Genome Information Analysis Core (SignAC) in ASHBi for their technical support. We are grateful to Dr. Hitoshi Niwa (Kumamoto University) for providing the pPB-CAG-cHA-pA, pCAGGS and pBRBlue II vectors and to Dr. Kazuo Yamagata (Kindai University) for providing the pcDNA3.1-H2B-mCherry-poly(A83) vector[38]. pCMV-hyPBase was provided by the Sanger Institute. This study was supported by JSPS KAKENHI Grant Numbers JP20H05763, JP20B302, JP20K21370, JP21H05038, JP22H02529, JP22K19246, JP23K23794 and JP24K21915 to Tomoyuki Tsukiyama; by a grant from the World Premier International Research Center Initiative (WPI) to M.N. and Tomoyuki Tsukiyama; by a grant from the Takeda Science Foundation to Tomoyuki Tsukiyama; by a grant from the Mochida Memorial Foundation for Medical and Pharmaceutical Research to Tomoyuki Tsukiyama; and by an in-house grant from Shiga University of Medical Science to M.N. and Tomoyuki Tsukiyama.

## Author contributions

M.N. performed the experiments, analyzed and interpreted the data, and wrote the paper. C.I., I.T., I.K., and H.T. generated the animals. Setsuko Tsukiyama-Fujii performed the experiments. A.M., Shoko Tarumoto, Taro Tsujimura and T.Y. conducted identification of genome insertion sites using NGS. T.I., Tomonori Nakamura and M.S. interpreted the data. Takahiro Nakagawa and I.I. conducted phenotypic analysis of the delivered monkey. Tomoyuki Tsukiyama conceived, designed, and performed the experiments, analyzed and interpreted the data, and wrote the paper.

## Competing interests

The authors declare no competing interests.
