## [Transparent Peer Review file · Nature Communications]

Non-viral generation of transgenic non-human primates via the piggyBac transposon system.

Corresponding Author: Dr Tomoyuki TSUKIYAMA

Version 1:

Reviewer comments:

Reviewer #1

(Remarks to the Author)

"Non-viral generation of transgenic non-human primates via the piggyBac transposon system." by Masataka Nakaya and colleagues.

In contrast to gene knockouts, there are significant problems with transgenesis (insertion and expression of additional DNA) in large animal models, including non-human primates. The most important of these is possibly transgene silencing. Lentiviral vectors, which have been mainly used for transgenes in NHPs, show an inconsistent expression pattern, which severely impairs the generation of good disease models or reporter lines, e.g. for analyzing brain function of NHPs. Therefore, testing alternative strategies for transgenesis of NHPs is an important goal. Retrotransposons, and in particular the PiggyBac transposon, are promising candidates for a better (more stable and reproducible) transgenesis of NHPs.

In the present manuscript, Masataka Nakaya and colleagues generated transgenic cynomolgus macaques (*Macaca fascicularis*) by piggyBac transposition in embryos. Three transgenic offspring were generated by co-injection of sperm, piggyBac vector and PBase mRNA into mature oocytes. Two of the three transgenic progeny died and were therefore available for more detailed molecular analysis of transgene expression characteristics and integration sites. The advantage of the PiggyBac transposon system is that it can transfer much larger DNA fragments compared to lentiviral vectors, which is very useful for generating transgenic NHPs harboring complex transgenes, large ORFs or lncRNAs. In addition, data from non-primate species suggest that transposons enable robust transgene expression with limited (epigenetic) transgene silencing.

Against this background and in view of the importance of NHP models in various areas of biomedicine, this study addresses an important question. Transgenic cells (genomes) were detected in all samples of the transgenic animals examined, suggesting broad and relatively robust expression of the transgenes. However, the data also show clear variability between the different tissues analyzed in this study. The transgene used in this study to establish the methods was elegantly designed. The manuscript is well written and the figures are of excellent quality and very detailed. The quality of the data is very good and generally supports the conclusions. Overall, this is an interesting study. In my view, however, some relevant questions still need to be clarified in order to realize the full potential of this study. Currently, the potential significance of the findings for the field is limited.

Specific comments:

1. The current manuscript investigates only the potential founder animals. It is known that transgenes can be silenced during germline transmission so that F1 animals show an altered transgene expression pattern compared to their parents. Moreover, an analysis of transgene segregation during breeding would be informative. It would therefore be important to also analyze F1 animals to draw meaningful conclusions about the robustness of the piggyBac system in NHPs - especially for the derivation of well-characterized transgenic NHP lines. (I am aware that this would require time-consuming additional experimentation / breeding, but it would make the manuscript much stronger if germline transmission and [whole body] transgene activity in F1 animals could be shown.)
2. As the transgenic animals have multiple insertion sites, it would be highly desirable to check whether all transgenes show comparable expression levels or whether there are significant differences between them regarding expression levels and (cell type-specific) expression patterns (depending on the insertion site). This could also be tested by breeding with wildtype animals, where the offspring would have only a reduced number / spectrum of copies of the transgene (ideally only

- one). As a first step it would be interesting to test the integration sites in germ cells / gametes of the surviving transgenic animal. Would this be possible?
3. As discussed in the manuscript, it is surprising / interesting that the animals show a high degree of mosaicism and that particularly the placenta shows clearly different integration patterns compared with the other organs. This strongly suggests that the integration (transposition) occurs surprisingly late during embryonic development, when trophoblast and embryoblast have segregated. Have the authors considered using PBase protein instead of mRNA to prevent / reduce unwanted mosaicism? Could TRIM-away be an approach (PMID: 29153837)?
 4. What are the integration sites of the transgenes? It would be interesting and should be possible to show them as the transgenic genomes have been sequenced. This would be interesting for two reasons: 1. Do the integration sites explain the death of animal #4? 2. The PiggyBac transposon has a tendency (at least in some cell types) to integrate at functionally relevant sites (PMID 20606646). Was this also the case in the transgenic monkeys?
 5. In those tissues, where only a subset of cells shows fluorescence, it would be interesting to see whether the promoter driving the transgene is methylated or not.
 6. In summary of these main points, the piggyBac approach (surprisingly) does not seem to be clearly superior to the lentiviral approach. This is an unexpected finding as transposons, particularly piggyBac and Sleeping Beauty, are thought to be (more) resistant against epigenetic silencing in vivo (than lentiviral transgenes) (e.g. PMID: 38918814, PMID: 29792157).
 7. The manuscript is sometimes a bit insufficiently referenced (e.g. first paragraph of the introduction and page 4 lines 27-30).
 8. How do you explain the discrepancy between Fig. 5b and Supp. Fig.4 (transcript detection in liver, spleen and testis of animal #1 but almost no fluorescence signal in the respective organs of this animal)?
 9. Supp. Fig.4 and 5 show no fluorescence for liver, spleen and testis in both animals, #1 and #4, while all other organs show fluorescence in both animals. Is there a reason why these organs do not fluoresce reproducibly?
 10. Page 5, line 22: Fig. 2b
 11. Page 6, line 29-30: Is it possible that also immune cells (which are presumably present in all organs) contribute to the finding that the transgenes were found in all tissue samples? Was the integration pattern in bone marrow analyzed? If so, could the respective "bone marrow bands" also be found in the other tissues?
 12. Page 7, line 14: The monkeys #1 and #4 died or were stillborn. Were there any pathological findings in these animals?
 13. A clear overview of the numbers of animals, oocytes (per laparoscopy), embryos, embryo transfers etc. would be very helpful in order to calculate the efficiencies of all the experimental steps.
 14. How were the surrogate mothers (pre-) treated and their reproductive cycles monitored? More information about the animal procedures would be appreciated.
 15. Fig. 5c: There is a discordance between EGFP (present), tdTomato and 5'ITR (both undetectable) in blood of animal #3? How do you explain this?
 16. Supp. Figure 1: please provide units.

Reviewer #2

(Remarks to the Author)

The manuscript discusses the generation of transgenic monkeys using the non-viral piggyBac transposon system. Traditionally, lentiviral vector expression in fertilized monkey eggs has been an established method for creating transgenic monkeys that express various transgenes. However, the lentiviral system has several shortcomings, which the non-viral piggyBac transposon system can potentially overcome.

The transgene expression via the piggyBac transposon in these transgenic monkeys is promising. However, more substantial evidence is needed to support the relative expression levels of the transgenes. The authors used low magnification micrographs to demonstrate transgene expression. In Figure 6, the relative expression rates of the transgene, examined by the percentage of positive cells at high magnification (Figures a-c), do not align with the quantitative bar graphs in Figure 6e. The expression data of the transgene requires more robust evidence. It would also be beneficial to use Western blotting to verify transgene expression, especially in germline cells.

The genomic integration sites and copy numbers of the transgene via the piggyBac transposon apparently influence transgene expression. It also seems that using the piggyBac transposon system results in a low yield of transgenic monkeys, possibly due to extensive genomic integration. The significant concerns and limitations of the piggyBac transposon system need to be compared with those of the lentiviral vector expression and thoroughly discussed.

Additional Considerations:

How many recipient female monkeys were used for the transfer of a total of 20 embryos? This critical information should be clearly indicated in the results section.

Version 2:

Reviewer comments:

Reviewer #1

(Remarks to the Author)

Revision of the manuscript "Non-viral generation of transgenic non-human primates via the piggyBac transposon system." by Masataka Nakaya and colleagues.

The authors have addressed my points of criticism and questions very satisfactorily for the most part. Thank you. I only have two points that I would like to ask the authors to consider:

1. In the rebuttal letter, the authors write that immune cells are no longer present in skin fibroblast cultures. At least in primary fibroblast cultures, immune cells may still be present (PMID: 33673402).
2. It would be of great interest to be able to analyze the F1 and ideally also the F2 generations of the founders. It is clear that this is not possible within the scope of this manuscript. Therefore, the approach of analyzing the question of transgene activity after germline transmission in a mammalian surrogate model, i.e., the mouse, is appreciated. However, there is some evidence that the same transgene behaves differently in mice and primates (PMID: 33673402; PMID: 11786607). Therefore, it should not be assumed with certainty that the findings in the mouse can be transferred 1:1 to the primate.

Reviewer #2

(Remarks to the Author)

The revision has addressed some of my concerns, but it also highlights additional weaknesses of the piggyBac system. One issue is the extensive integration of the transgene, which may result in a low yield of newborn monkeys. Another is the low expression level of the transgene in the brain, likely due to variability in the extent of gene integration and differing levels of transgene expression. Despite these limitations, I believe this study represents the first successful demonstration of live transgenic monkeys generated using the piggyBac transposon system that is able to express a large transgene. However, in the discussion, the authors should explicitly address the limitations of their study. Clearly outlining these weaknesses will help other researchers determine whether this transgenic approach is suitable for their specific research needs.

Thank you very much for your helpful suggestions regarding our manuscript. We have attempted to make all necessary changes to address the concerns of the reviewers. Point-by-point responses are shown below.

Comments of the Reviewers:

Reviewer #1:

"Non-viral generation of transgenic non-human primates via the piggyBac transposon system." by Masataka Nakaya and colleagues.

In contrast to gene knockouts, there are significant problems with transgenesis (insertion and expression of additional DNA) in large animal models, including non-human primates. The most important of these is possibly transgene silencing. Lentiviral vectors, which have been mainly used for transgenes in NHPs, show an inconsistent expression pattern, which severely impairs the generation of good disease models or reporter lines, e.g. for analyzing brain function of NHPs. Therefore, testing alternative strategies for transgenesis of NHPs is an important goal. Retrotransposons, and in particular the PiggyBac transposon, are promising candidates for a better (more stable and reproducible) transgenesis of NHPs.

*In the present manuscript, Masataka Nakaya and colleagues generated transgenic cynomolgus macaques (*Macaca fascicularis*) by piggyBac transposition in embryos. Three transgenic offspring were generated by co-injection of sperm, piggyBac vector and PBase mRNA into mature oocytes. Two of the three transgenic progeny died and were therefore available for more detailed molecular analysis of transgene expression characteristics and integration sites. The advantage of the PiggyBac transposon system is that it can transfer much larger DNA fragments compared to lentiviral vectors, which is very useful for generating transgenic NHPs harboring complex transgenes, large ORFs or lncRNAs. In addition, data from non-primate species suggest that transposons enable robust transgene expression with limited (epigenetic) transgene silencing.*

Against this background and in view of the importance of NHP models in various areas of biomedicine, this study addresses an important question. Transgenic cells (genomes) were detected in all samples of the transgenic animals examined, suggesting broad and

relatively robust expression of the transgenes. However, the data also show clear variability between the different tissues analyzed in this study. The transgene used in this study to establish the methods was elegantly designed. The manuscript is well written and the figures are of excellent quality and very detailed. The quality of the data is very good and generally supports the conclusions. Overall, this is an interesting study. In my view, however, some relevant questions still need to be clarified in order to realize the full potential of this study. Currently, the potential significance of the findings for the field is limited.

Thank you very much for your positive evaluations and insightful comments on our manuscript. The items that needed to be addressed in detail are explained in the point-by-point responses below.

Specific comments:

1. The current manuscript investigates only the potential founder animals. It is known that transgenes can be silenced during germline transmission so that F1 animals show an altered transgene expression pattern compared to their parents. Moreover, an analysis of transgene segregation during breeding would be informative. It would therefore be important to also analyze F1 animals to draw meaningful conclusions about the robustness of the piggyBac system in NHPs - especially for the derivation of well-characterized transgenic NHP lines. (I am aware that this would require time-consuming additional experimentation / breeding, but it would make the manuscript much stronger if germline transmission and [whole body] transgene activity in F1 animals could be shown.)

We appreciate your valuable suggestion regarding the importance of F1 analysis and transgene segregation to evaluate the robustness of the piggyBac system in NHPs. We acknowledge that germline transmission and whole-body transgene activity in F1 animals would provide valuable insights. However, as cynomolgus monkeys require 4–5 years to reach sexual maturity and become capable of breeding, such analysis is currently unfeasible within the timeframe of this study.

To address this limitation, we conducted germline transmission studies using the same piggyBac vector in mice with a significantly shorter reproductive cycle. In these experiments, F1 mice generated by mating a founder (F0) male with C57BL/6 females exhibited a transgene transmission rate of 72.2% (13/18). One F1 individual carried EGFP but not tdTomato. Repeat PCR confirmed the presence of EGFP, with detection levels comparable to other positive individuals, suggesting that contamination was unlikely. This result was likely due to the random integration of a partial vector sequence. Nevertheless, this was an infrequent phenomenon (1/13) and did not compromise the overall practicality of the system. Additionally, ICSI experiments using sperm from three different F1 male mice resulted in high fluorescence-positive embryo rates of 85.7%, 72.7%, and 54.5%, demonstrating efficient germline transmission. These results align with previous studies (PMID: 38918814; PMID: 29792157) reporting that the piggyBac system is relatively resistant to epigenetic silencing, in contrast to lentiviral methods. Although direct F1 data in NHPs are unavailable, we believe these results support the robustness and practicality of the piggyBac system for generating transgenic lines. We added this information on P. 5, L. 29–P. 6, L. 6.

2. As the transgenic animals have multiple insertions sites, it would be highly desirable to check whether all transgenes show comparable expression levels or whether there are significant differences between them regarding expression levels and (cell type-specific) expression patterns (depending on the insertion site). This could also be tested by breeding with wildtype animals, where the offspring would have only a reduced number / spectrum of copies of the transgene (ideally only one). As a first step it would be interesting to test the integration sites in germ cells / gametes of the surviving transgenic animal. Would this be possible?

Thank you for your insightful suggestion. We agree that examining the expression levels and patterns of individual insertions would provide valuable insights. However, our current study does not utilize molecular barcoding, making it challenging to analyze each insertion site specifically through a transcriptomic approach. However, we sorted fibroblasts from Monkey #1 into fluorescence-positive and -negative populations and observed that some insertion sites were detectable even in fluorescence-negative cells.

This suggests that only a subset of the insertions contributes to actual transgene expression.

While analyzing germ cells or gametes from the surviving transgenic animal would be informative, it is important to note that this monkey, monkey #2, would require an additional 4–5 years to reach sexual maturity. Monkey #2 had an average copy number of approximately two yet displayed robust fluorescence, and therefore we believe the number of functionally contributing copies was likely one or two. Additionally, in our F1 germ cell analysis in mice we did not observe any obvious evidence of piggyBac silencing over generations, supporting the stability of its expression. Even if position effects result in only one copy contributing to fluorescence expression, we expect at least 50% of offspring in the next generation to exhibit fluorescence expression, which we consider a practical and promising outcome.

3. As discussed in the manuscript, it is surprising / interesting that the animals show a high degree of mosaicism and that particularly the placenta shows clearly different integration patterns compared with the other organs. This strongly suggests that the integration (transposition) occurs surprisingly late during embryonic development, when trophoblast and embryoblast have segregated. Have the authors considered using PBase protein instead of mRNA to prevent / reduce unwanted mosaicism? Could TRIM-away be an approach (PMID: 29153837)?

We appreciate your insightful comment regarding the potential use of PBase protein to mitigate mosaicism. We have indeed explored an approach involving the addition of a drug-dependent destabilization signal to PBase and protein injection, which demonstrated a reduction in mosaicism. While promising, this approach falls outside the primary scope of the current study, which focuses on generating transgenic monkeys using piggyBac transposition. We plan to present these findings in a separate publication.

4. What are the integration sites of the transgenes? It would be interesting and should be possible to show them as the transgenic genomes have been sequenced. This would be interesting for two reasons: 1. Do the integration sites explain the death of animal #4? 2.

The PiggyBac transposon has a tendency (at least in some cell types) to integrate at functionally relevant sites (PMID 20606646). Was this also the case in the transgenic monkeys?

Thank you for raising this important point. We conducted an analysis of the integration sites for the transgenes in all four monkeys. A total of 30 integration sites were identified, with the following distribution: 15 in intergenic regions, 9 in introns, 4 upstream of genes, 1 in a UTR, and 1 in a coding region. This general pattern was consistent across the animals, including the two that did not survive (monkeys #1 and #4). Interestingly, we observed upstream integrations in both monkey #1 (MRPL48, WDR19) and monkey #4 (TLCD5), but their specific effects on development remain unclear.

Regarding the integration tendencies of piggyBac, we noted a difference from the observations of Huang et al. (PMID 20606646), who reported a high preference for insertion within 5 kb of transcription start sites (TSSs). In contrast, most of the integration sites in our monkeys were located outside this range. This raises the possibility that successful term development might be linked to an avoidance of transgene integration into critical genomic loci. We added this information on P. 8, L. 12–25.

5. In those tissues, where only a subset of cells shows fluorescence, it would be interesting to see whether the promoter driving the transgene is methylated or not.

We appreciate this recommendation, and agree that an exploration of the relationship between transgene mosaicism and DNA methylation could be helpful. We therefore conducted bisulfite PCR and analyzed the methylation status at the CAG promoter of the transgene in fibroblast subpopulations with strongly positive and negative fluorescence. Unexpectedly, no significant differences in methylation levels were observed between these fractions. We then further analyzed tissues with varying levels of transgene expression, including the brain and liver (low expression) and the heart and skeletal muscle (high expression). While a trend toward hypomethylation was observed in the heart, no statistically significant differences were found. These findings suggest that transgene mosaicism cannot be fully explained by DNA methylation alone and may involve other epigenetic regulatory mechanisms, such as histone modifications or

chromatin structure alterations. We added this information on P. 9, L. 28–P. 10, L. 5.

6. In summary of these main points, the piggyBac approach (surprisingly) does not seem to be clearly superior to the lentiviral approach. This is an unexpected finding as transposons, particularly piggyBac and Sleeping Beauty, are thought to be (more) resistant against epigenetic silencing in vivo (than lentiviral transgenes) (e.g. PMID: 38918814, PMID: 29792157).

Thank you for drawing our attention to the differential silencing susceptibility between piggyBac and lentiviral transgenes. Previous studies using lentiviral vectors have reported significant silencing attributed to global hypermethylation of the transgene promoter. In contrast, our DNA methylation analysis of the CAG promoter in the piggyBac transgenic monkey demonstrated mixes of hypomethylated and hypermethylated sequences within the same tissues. This suggests that position effects, rather than piggyBac-specific methylation, are the dominant factor influencing DNA methylation. Additionally, our germline transmission analysis in mice revealed robust fluorescence in a substantial proportion of F2 embryos. This is in stark contrast to a previous study using lentiviral vectors, where increased DNA methylation leading to silencing over generations was observed. Our findings align with previous reports (e.g., PMID: 38918814; PMID: 29792157) that highlight piggyBac's greater resistance to epigenetic silencing compared to lentiviral transgenes. We have added a discussion of this topic to the manuscript on P. 11, L. 31–P. 12, L. 9.

7. The manuscript is sometimes a bit insufficiently referenced (e.g. first paragraph of the introduction and page 4 lines 27-30).

We appreciate your feedback on the need for additional references. To address this, we have added appropriate citations to the first paragraph of the introduction. The statement regarding green nuclear autofluorescence in monkey blastocyst embryos was based on our observations during previous research in monkey developmental biology. To lend support to this statement, we have added Supplementary Fig. 1, which

illustrates this phenomenon. We then added this information to the main text on P. 4, L. 27–29.

8. How do you explain the discrepancy between Fig. 5b and Supp. Fig.4 (transcript detection in liver, spleen and testis of animal #1 but almost no fluorescence signal in the respective organs of this animal)?

Thank you for pointing out the difference between qPCR results and the fluorescence images. The discrepancy can be attributed to differences in the sensitivity and resolution of the methods used. While qPCR analysis confirmed transgene expression in all examined tissues, fluorescence imaging at low magnification did not consistently reveal strong signals in some organs, such as the liver, spleen, lung, kidneys and testis. However, subsequent immunostaining and high-magnification imaging demonstrated the presence of fluorescent signals in these tissues, though only in a low proportion of cells with weak fluorescence intensity. This limited cell positivity and low fluorescence intensity likely explains the weak or undetectable fluorescence in the initial low-magnification images. We have added several brief passages to clarify the discrepancy on P. 9, L. 9–15 and P. 10, L. 12–14.

9. Supp. Fig.4 and 5 show no fluorescence for liver, spleen and testis in both animals, #1 and #4, while all other organs show fluorescence in both animals. Is there a reason why these organs do not fluoresce reproducibly?

We appreciate your comment regarding the reproducibility of fluorescence detection in these tissues. Both figures present low-magnification fluorescence images, which are less sensitive for detecting signals when the proportion of fluorescently positive cells is low. As previously noted, high-magnification immunostaining confirmed the presence of fluorescence in these tissues, suggesting that the apparent lack of fluorescence in low-magnification images was due to the limitations of this imaging technique. We have clarified this point in the manuscript on P. 9, L. 9–15.

10. Page 5, line 22: Fig. 2b

Thank you very much. We corrected the description.

11. Page 6, line 29-30: *Is it possible that also immune cells (which are presumably present in all organs) contribute to the finding that the transgenes were found in all tissue samples? Was the integration pattern in bone marrow analyzed? If so, could the respective “bone marrow bands” also be found in the other tissues?*

Thank you for raising this important point. We agree that immune cells, which are present across all tissues, could potentially contribute to the detection of transgenes in genomic PCR analysis. Although a detailed integration analysis of bone marrow has not been conducted, our immunostaining results demonstrate transgene expression in tissue-specific cells that are morphologically distinct from immune cells. Additionally, cultured fibroblasts, which do not contain immune cells, also exhibited transgene expression. This suggests that the detected transgene is not solely derived from immune cells. To address this, we have revised the manuscript to clarify that immunofluorescence staining was performed to confirm transgene expression at the single-cell level, specifically to distinguish immune cell-derived from tissue-specific transgene expression. This revision provides a clear rationale for the use of immunostaining in our study. We added this information on P. 8, L. 27–P. 9, L. 5.

12. Page 7, line 14: *The monkeys #1 and #4 died or were stillborn. Were there any pathological findings in these animals?*

We appreciate your inquiry concerning potential pathological findings in monkeys #1 and #4. Comprehensive necropsy examinations were performed, but no pathological abnormalities were observed. We have added this information to the manuscript for completeness on P. 7, L. 9–10.

13. A clear overview of the numbers of animals, oocytes (per laparoscopy), embryos, embryo transfers etc. would be very helpful in order to calculate the efficiencies of all the experimental steps.

We appreciate your suggestion. We have updated the relevant table to include the total number of oocytes collected during each retrieval procedure. Additionally, we have clarified in the figure legend that "ET" refers to "embryo transfer." Since only one animal was used per laparoscopy, we opted to include this detail in the figure legend rather than the table in order to maintain concision of the table. We have also updated the table related to transgenic mouse generation with the number of oocytes used and the number of blastocysts developed.

14. How were the surrogate mothers (pre-) treated and their reproductive cycles monitored? More information about the animal procedures would be appreciated.

Thank you for the suggestion. We monitored the menstrual cycles of all female monkeys daily. Based on these records, we identified potential recipient females whose reproductive cycles were optimal for embryo transfer. The final selection of surrogate mothers was determined through laparoscopic examination of follicular development and ovulation scars to ensure the best possible synchronization for embryo implantation. We have added these details to the manuscript for clarity on P. 17, L. 5–9.

15. Fig. 5c: There is a discordance between EGFP (present), tdTomato and 5'ITR (both undetectable) in blood of animal #3? How do you explain this?

Thank you for highlighting the discordance in Figure 5c. To address this, we conducted additional qPCR experiments using newly designed primer and probe sets targeting various regions of EGFP. These analyses revealed that EGFP signals were detectable in Tg #3 samples, but at much lower levels than in Tg #2, which was definitively identified as a positive transgenic animal.

Moreover, although EGFP signals were detected across its entire coding sequence (from the N-terminal to C-terminal) in Tg #3, no amplification was observed at the H2B-EGFP junction region. As it is highly improbable that EGFP was inserted cleanly and exclusively without its flanking sequences, these findings indicate that the detected EGFP signals in Tg #3 result from trace contamination rather than partial genomic integration.

Additionally, in our ddPCR analysis, we observed non-zero EGFP detection even in WT samples with 200 ng of DNA. This suggests that the EGFP ddPCR primer in the original analysis may have been overly sensitive, potentially amplifying minimal contamination or leading to the erroneous recognition of homologous sequences. Importantly, we have rigorously validated the integration sites and surrounding cassettes for confirmed transgenic animals, ensuring that this issue does not compromise the central conclusions of our study.

To reflect these findings, we have revised the manuscript to remove suggestions of partial EGFP insertion and to clarify the possibility of contamination. We added this information on P. 7, L. 27–33.

16. Supp. Figure 1: please provide units.

Thank you for pointing this out. We have updated the figure legend with the units of concentration (ng/μL) for both the PB vector and PBase.

Reviewer #2:

The manuscript discusses the generation of transgenic monkeys using the non-viral piggyBac transposon system. Traditionally, lentiviral vector expression in fertilized monkey eggs has been an established method for creating transgenic monkeys that express various transgenes. However, the lentiviral system has several shortcomings, which the non-viral piggyBac transposon system can potentially overcome.

The transgene expression via the piggyBac transposon in these transgenic monkeys is promising. However, more substantial evidence is needed to support the relative expression levels of the transgenes.

We appreciate your comments and useful suggestions, which helped us improve our paper. The items that need to be addressed in detail are explained in the point-by-point responses below.

The authors used low magnification micrographs to demonstrate transgene expression. In Figure 6, the relative expression rates of the transgene, examined by the percentage of positive cells at high magnification (Figures a-c), do not align with the quantitative bar graphs in Figure 6e. The expression data of the transgene requires more robust evidence. It would also be beneficial to use Western blotting to verify transgene expression, especially in germline cells.

We greatly appreciate this insightful comment; it helped us clarify key aspects of our analysis. The high-magnification micrographs shown are representative images, whereas the quantitative bar graphs are derived from the analysis of 10 high-magnification fields per sample. To ensure clarity, we added a detailed explanation of the protocol of our quantitative analysis to the figure legend. To further substantiate our findings, we performed Western blot analysis for transgene expression in multiple tissues, including testis tissue. As shown in Figure 5c, transgene expression was detected in all tissues analyzed. We hope this additional evidence addresses your concerns. We added this information on P. 7, L. 14–17.

The genomic integration sites and copy numbers of the transgene via the piggyBac transposon apparently influence transgene expression. It also seems that using the piggyBac transposon system results in a low yield of transgenic monkeys, possibly due to extensive genomic integration. The significant concerns and limitations of the piggyBac transposon system need to be compared with those of the lentiviral vector expression and thoroughly discussed.

We appreciate your constructive feedback regarding the limitations and potential risks associated with the piggyBac transposon system. Both piggyBac and lentiviral vectors share the limitation of random integration, which may result in insertional mutagenesis or embryotoxicity due to excessive genomic insertions. This shared limitation highlights the need for advancements in targeted integration techniques. However, the current efficiency of knock-in approaches, particularly for large transgene fragments, remains suboptimal. Despite these challenges, piggyBac offers distinct advantages over lentiviral vectors. Specifically, piggyBac can efficiently accommodate and integrate larger DNA fragments, and its non-viral nature minimizes biosafety concerns. Furthermore, piggyBac facilitates preimplantation selection of transgenic embryos, which is particularly advantageous in primate research. To address these points comprehensively, we have expanded the manuscript to include a balanced discussion comparing the strengths and limitations of both systems. We added this information on P. 12, L. 33–P. 13, L. 9.

Additional Considerations:

*How many recipient female monkeys were used for the transfer of a total of 20 embryos?
This critical information should be clearly indicated in the results section.*

We agree. Knowledge of the number of recipient female monkeys used is crucial for understanding the experimental design. We added a sentence in the Results section stating that one embryo was transferred into each recipient female, consistent with the information provided in the Methods section. This revision ensures that readers can readily understand the experimental design without needing to refer back to the Methods.

We sincerely appreciate the insightful comments and suggestions provided by the reviewers. In this second revision, we have thoroughly incorporated the necessary changes to address their concerns. Our detailed responses to each point are included below. Thank you for the opportunity to improve our manuscript further.

Comments of the Reviewers:

Reviewer #1:

Revision of the manuscript "Non-viral generation of transgenic non-human primates via the piggyBac transposon system." by Masataka Nakaya and colleagues.

The authors have addressed my points of criticism and questions very satisfactorily for the most part. Thank you. I only have two points that I would like to ask the authors to consider:

1. In the rebuttal letter, the authors write that immune cells are no longer present in skin fibroblast cultures. At least in primary fibroblast cultures, immune cells may still be present (PMID: 33673402).

We appreciate your comment regarding the presence of immune cells in primary fibroblast cultures. We have withdrawn the corresponding statement mentioned in our rebuttal letter, and we confirm that it is not included in the revised manuscript. Nevertheless, our principal conclusion—that positive cells are present beyond the immune cell population—is still firmly supported by immunostaining data from other tissues.

2. It would be of great interest to be able to analyze the F1 and ideally also the F2 generations of the founders. It is clear that this is not possible within the scope of this manuscript. Therefore, the approach of analyzing the question of transgene activity after germline transmission in a mammalian surrogate model, i.e., the mouse, is appreciated. However, there is some evidence that the same transgene behaves differently in mice and primates (PMID: 33673402; PMID: 11786607). Therefore, it should not be assumed with certainty that the findings in the mouse can be transferred 1:1 to the primate.

We appreciate your insight regarding the potential variability in transgene behavior between species. Accordingly, we have modified the Discussion section to include the possibility that the transgene might behave differently in mice and primates. This addition provides important context for the interpretation of our results.

Reviewer #2:

The revision has addressed some of my concerns, but it also highlights additional weaknesses of the piggyBac system. One issue is the extensive integration of the transgene, which may result in a low yield of newborn monkeys. Another is the low expression level of the transgene in the brain, likely due to variability in the extent of gene integration and differing levels of transgene expression. Despite these limitations, I believe this study represents the first successful demonstration of live transgenic monkeys generated using the piggyBac transposon system that is able to express a large transgene. However, in the discussion, the authors should explicitly address the limitations of their study. Clearly outlining these weaknesses will help other researchers determine whether this transgenic approach is suitable for their specific research needs.

We appreciate your comment regarding limitations in the piggyBac system. We have now amended the Discussion to include the opinion that, as evidenced by the low expression in the brain in our study, the promoter choice can contribute to variability in transgene expression. This clarification enhances the discussion of vector limitations and the necessity for improved targeting techniques.